# AdaRL: What, Where, and How to Adapt in Transfer Reinforcement Learning

**Biwei Huang**
Carnegie Mellon University
biweih@andrew.cmu.edu

**Fan Feng**
City University of Hong Kong
ffeng1017@gmail.com

**Chaochao Lu**
University of Cambridge & Max Planck Institute for Intelligent Systems
cl641@cam.ac.uk

**Sara Magliacane**
University of Amsterdam & MIT-IBM Watson AI Lab
sara.magliacane@gmail.com

**Kun Zhang**
Carnegie Mellon University &
Mohamed bin Zayed University of Artificial Intelligence
kunz1@cmu.edu

## Abstract

One practical challenge in reinforcement learning (RL) is how to make quick adaptations when faced with new environments. In this paper, we propose a principled framework for adaptive RL, called *AdaRL*, that adapts reliably and efficiently to changes across domains with a few samples from the target domain, even in partially observable environments. Specifically, we leverage a parsimonious graphical representation that characterizes structural relationships over variables in the RL system. Such graphical representations provide a compact way to encode what and where the changes across domains are, and furthermore inform us with a minimal set of changes that one has to consider for the purpose of policy adaptation. We show that by explicitly leveraging this compact representation to encode changes, we can efficiently adapt the policy to the target domain, in which only a few samples are needed and further policy optimization is avoided. We illustrate the efficacy of AdaRL through a series of experiments that vary factors in the observation, transition and reward functions for Cartpole and Atari games [1].

## 1 Introduction and Related Work

Over the last decades, reinforcement learning (RL) (Sutton and Barto, 1998) has been successful in many tasks (Mnih et al., 2013; Silver et al., 2016). Most of these early successes focus on a fixed task in a fixed environment. However, in real applications we often have changing environments, and it has been demonstrated that the optimal policy learned in a specific domain may not be generalized to other domains (Taylor and Stone, 2009). In contrast, humans are usually good at transferring acquired knowledge to new environments and tasks both efficiently and effectively (Pearl and Mackenzie, 2018), thanks to the ability to understand the environments. Generally speaking, to achieve reliable, low-cost, and interpretable transfer, it is essential to understand the underlying process—which decision-making factors have changes, where the changes are, and how they change, instead of transferring blindly (e.g., transferring the distribution of high-dimensional images directly).

There are roughly two research lines in transfer RL (Taylor and Stone, 2009; Zhu et al., 2020): (1) finding policies that are robust to environment variations, and (2) adapting policies from the source domain to the target domain as efficiently as possible. For the first line, the focus is on learning policies that are robust to environment variations, e.g., by maximizing a risk-sensitive objective over a distribution of environments (Tamar et al., 2015) or by extracting a set of invariant states (Zhang et al., 2020a; 2021a; Tomar et al., 2021). A more recent method encodes task-relevant invariances by putting behaviorally equivalent states together, which helps better generalization (Agarwal et al., 2021). On the other hand, with the increase of the number of domains, the common part may get

---

[1]Code link: https://github.com/Adaptive-RL/AdaRL-code

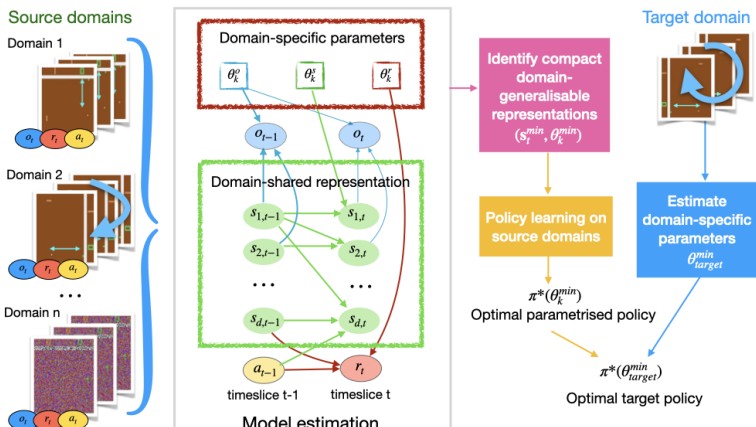

Figure 1: The overall AdaRL framework. We learn a Dynamic Bayesian Network (DBN) over the observations, latent states, reward, actions and domain-specific change factors that is shared across the domains. We then characterize a minimal set of representations that suffice for policy transfer, so that we can quickly adapt the optimal source policy with only a few samples from the target domain.

even smaller, running counter to the intention of collecting more information with more domains. Moreover, focusing only on the invariant part and disregarding domain-specific information may not be optimal; for instance, in the context of domain adaptation, it has been demonstrated that the variable part also contains information helpful to improve prediction accuracy (Zhang et al., 2020b).

In this paper, we propose a method along the second line, adapting source policies to the target. Approaches along this line adapt knowledge from source domains and reuse it in the target domain to improve data efficiency, i.e., in order for the agent to require fewer explorations to learn the target policy. For example, an agent could use importance reweighting on samples $\langle s, a, r, s' \rangle$ from sources (Tirinzoni et al., 2018; 2019) or start from the optimal source policy to initialize a learner in the target domain, as a near-optimal initializer (Taylor et al., 2007; Fernández et al., 2010). Another widely-used technique is finetuning: a model is pretrained on a source domain and the output layers are finetuned via backpropagation in the target domain (Hinton and Salakhutdinov, 2006; Mesnil et al., 2012). PNNs (Rusu et al., 2016), instead, retain a pool of pretrained models and learn lateral connections from them to extract useful features for a new task. Moreover, a set of approaches focus on sim2real transfer by adapting the parameters (Yu et al., 2017; Peng et al., 2020). However, many of these approaches still require a large amount of explorations and optimization in the target domain.

Recently, meta-RL approches such as MAML (Finn et al., 2017), PEARL (Rakelly et al., 2019), CAVIA (Zintgraf et al., 2019), Meta-Q learning (Fakoor et al., 2020), and others (Mendonca et al., 2019; Clavera et al., 2019; Duan et al., 2017) have been successfully applied to learn an inductive bias that accelerates the learning of a new task by training on a large number of tasks. Some of these methods (e.g., CAVIA and PEARL), as well as some prior work (e.g., HiMDPs (Doshi-Velez and Konidaris, 2016)) and recent follow-ups (Zhang et al., 2021b), have a similar motivation to our work: in a new environment not all parameters need to be updated, so we can force the model to only adapt a set of *context parameters*. However, these methods mostly focus on MDPs (except the Block MDP assumption in Zhang et al. (2021b)) and model all changes as a black-box, which may be less efficient for adaptation, as opposed to a factorized representation of change factors.

Considering these limitations, we propose AdaRL, a transfer RL approach that achieves low-cost, reliable, and interpretable transfer for partially observable Markov decision processes (POMDPs), with MDPs as a special case. In contrast to state-of-the-art approaches, we learn a parsimonious graphical representation that is able to characterize structural relationships among different dimensions of states, change factors, the perception, the reward variable, and the action variable. It allows us to model changes in transition, observation, and reward functions in a component-wise way. This representation is related to Factored MDPs (Kearns and Koller, 1999; Boutilier et al., 2000; Strehl et al., 2007) and Factored POMDPs (Katt et al., 2019), but augmented with change factors that represent a low-dimensional embedding of the changes across domains. Our main motivation is that distribution shifts are usually localized – they are often due to the changes of only a few variables in the generative processes, so we can just adapt the distribution of a small portion of variables (Huang

et al., 2020; Schölkopf et al., 2021) and, furthermore, factorized according to the graph structure, each distribution module can be adapted separately (Schölkopf, 2019; Zhang et al., 2020b).

In Fig. 1 we give a motivating example and a general description of AdaRL. In this example, we consider learning policies for Pong (Bellemare et al., 2013) that can easily generalize to different rotations $\omega$ and to images corrupted with white noise. Specifically, given data from $n$ source domains with different rotations and noise variances, we learn a parsimonious latent state representation shared by all domains, denoted by $\mathbf{s}_t$, and characterize the changes across domains by a two-dimensional factor $\boldsymbol{\theta}_k$. We identify a set of minimal sufficient representations $(\mathbf{s}_t^{min}, \boldsymbol{\theta}_k^{min})$ for policy transfer. For instance, here only the rotation factor $\omega$ needs adapting (i.e., $\boldsymbol{\theta}_k^{min} = \omega_k$), since the noise factor does not affect the optimal policy. Similarly, as we will show formally in the rest of the paper, not all components $s_{i,t}$ of the state vector $\mathbf{s_t}$ are necessary for policy transfer. For example, $s_{2,t} \notin \mathbf{s}_t^{min}$, since it never affects the future reward. We learn an optimal policy $\pi^*(\cdot|\boldsymbol{\theta}_k^{min})$ on source domains. In the target domain, we only need a few samples to quickly estimate the value of the low-dimensional $\boldsymbol{\theta}_{\text{target}}^{min}$, and then we can apply $\pi^*(\cdot|\boldsymbol{\theta}_{target}^{min})$ directly. Our main contributions are summarized below:

- We assume a generative environment model, which explicitly takes into account the structural relationships among variables in the RL system. Such graphical representations provide a compact way to encode what and where the changes across domains are.
- Based on this model, we characterize a minimal set of representations that suffice for policy learning across domains, including the domain-specific change factors and domain-shared state representations. With this characterization, we adapt the policy with only a few target samples and without policy optimization in the target domain, achieving low-cost and reliable policy transfer.
- By leveraging a compact way to encode the changes, we also benefit from multi-task learning in model estimation. In particular, we propose the Multi-model Structured Sequential Variational Auto-Encoder (MiSS-VAE) for reliable model estimation in general cases.

## 2 A Compact Representation of Environmental Shifts

Suppose there are $n$ source domains and $n'$ target domains. In each source domain, we observe sequences $\{\langle o_t, a_t, r_t \rangle\}_{t=1}^T$, where $o_t \in \mathcal{O}$ are the perceived signals at time $t$ (e.g., images), $a_t \in \mathcal{A}$ is the executed action, and $r_t \in \mathcal{R}$ is the reward signal. We denote the underlying latent states by $\mathbf{s}_t = (s_{1,t}, \cdots, s_{d,t})^\top$, where $d$ is the dimensionality of latent states. We assume that the generative process of the environment in the $k$-th domain (with $k = 1, \ldots n + n'$) can be described in terms of the transition function for each dimension of $\mathbf{s}$ and the observation and reward functions as

$$\begin{cases} s_{i,t} &= f_i(\mathbf{c}_i^{\mathbf{s} \to \mathbf{s}} \odot \mathbf{s}_{t-1}, c_i^{a \to \mathbf{s}} \cdot a_{t-1}, \mathbf{c}_i^{\theta_k \to \mathbf{s}} \odot \boldsymbol{\theta}_k^{\mathbf{s}}, \epsilon_{i,t}^s), \text{ for } i = 1, \cdots, d, \\ o_t &= g(\mathbf{c}^{\mathbf{s} \to o} \odot \mathbf{s}_t, c^{\theta_k \to o} \cdot \theta_k^o, \epsilon_t^o), \\ r_t &= h(\mathbf{c}^{\mathbf{s} \to r} \odot \mathbf{s}_{t-1}, c^{a \to r} \cdot a_{t-1}, c^{\theta_k \to r} \cdot \theta_k^r, \epsilon_t^r), \end{cases} \tag{1}$$

where $\odot$ denotes the element-wise product, the $\epsilon_{i,t}^s, \epsilon_t^o, \epsilon_t^r$ terms are i.i.d. random noises. As explained below, $\mathbf{c}^{\cdot \to \cdot}$ are *masks* (binary vectors or scalars that represent structural relationships from one variable to the other), and $\boldsymbol{\theta_k} = (\theta_k^{\mathbf{s}}, \theta_k^o, \theta_k^r)$ are the change factors that have a constant value in each domain, but vary across domains in the transition, observation, and reward function, respectively. The latent states $\mathbf{s}_{t+1}$ form an MDP: given $\mathbf{s}_t$ and $a_t$, $\mathbf{s}_{t+1}$ is independent of previous states and actions. The perceived signals $o_t$ are generated from the underlying states $\mathbf{s}_t$. The actions $a_t$ directly influence the latent states $\mathbf{s}_{t+1}$, instead of the observed signals $o_t$, and the reward is determined by the latent states and the action. Eq. 1 can also represent MDPs as a special case if states $\mathbf{s_t}$ are directly observed, in which case the observation function of $o_t$ is not needed.

**Structural relationships and graphs.** Often the action variable $a_{t-1}$ does not influence every dimension of $\mathbf{s}_t$, and similarly, the reward $r_t$ may not be influenced by every dimension of $\mathbf{s}_{t-1}$. Furthermore, there are structural relationships between different dimensions of $\mathbf{s}_{t-1}$ and $\mathbf{s}_t$. To characterize these constraints, we explicitly take into account the graph structure $\mathcal{G}$ over the variables in the system characterized by a Dynamic Bayesian Network (Murphy, 2002) and encode the edges with masks $\mathbf{c}^{\cdot \to \cdot}$. In the first equation in Eq. 1 the transition function for the state component $s_i$, where the $j$th entry of $\mathbf{c}_i^{\mathbf{s} \to \mathbf{s}} \in \{0,1\}^d$ is 1 if and only if $s_{j,t}$ influences $s_{i,t+1}$ (graphically represented by an edge), while $c_i^{a \to \mathbf{s}} \in \{0,1\}$ is 1 if and only if the action $a_t$ has any effect on $s_{i,t+1}$. Similarly, the binary vector $\mathbf{c}_i^{\theta_k \to \mathbf{s}} \in \{0,1\}^p$ encodes which components of the change factor $\boldsymbol{\theta}_k^{\mathbf{s}} = (\theta_{1,k}^s, \ldots, \theta_{p,k}^s)^\top$ affect $s_{i,t+1}$. The masks in the observation function $g$ and reward function $h$ have similar functions. The

masks and the parameters of the functions $f$, $g$, and $h$, are invariant; all changes are encoded in $\boldsymbol{\theta}_k$. For simplicity of notation, we collect all the transition mask vectors in the matrices $\mathbf{C}^{\mathbf{s} \to \mathbf{s}} := [\mathbf{c}_i^{\mathbf{s} \to \mathbf{s}}]_{i=1}^d$ and $\mathbf{C}^{\theta_k \to \mathbf{s}} := [\mathbf{c}_i^{\theta_k \to \mathbf{s}}]_{i=1}^d$ and the scalars in the vector $\mathbf{c}^{a \to \mathbf{s}} := [c_i^{a \to \mathbf{s}}]_{i=1}^d$.

**Characterization of change factors in a compact way.** In practical scenarios, the environment model may change across domains. Moreover, it is often the case that given a high-dimensional input, only a few factors may change, which is known as *minimal change principle* (Ghassami et al., 2018) or *sparse mechanism shift assumption* (Schölkopf et al., 2021). In such a case, instead of learning the distribution shift over the high-dimensional input, thanks to the parsimonious graphical representation, we introduce a low-dimensional vector $\boldsymbol{\theta}_k$ to characterize the domain-specific information in a compact way (Zhang et al., 2020b). Specifically, $\theta_k^o$, $\theta_k^r$, and $\boldsymbol{\theta}_k^{\mathbf{s}}$ capture the change factors in the observation function, reward function, and transition dynamics, respectively; each of them can be multi-dimensional and that they are constant within each domain. In general, $\boldsymbol{\theta}_k$ can capture both the changes in the influencing strength and those in the graph structure, e.g., some edges may appear only in some domains. Since we assume that the structural relationships in Eq. 1 are invariant across domains, this means that the masks $\mathbf{c}^{\cdot \to \cdot}$ have to encode an edge even if it presents only in one domain, and furthermore, since $\boldsymbol{\theta}_k$ encodes the changes, it can switch the edge off in other domains. Fig. 1 shows an example of the graphical representation of the (estimated) environment model. Specifically, in this example, $\theta_k^s$ only influences $s_{1,t}$, $a_{t-1}$ does not have an edge to $s_{1,t}$, and among the states, only $s_{d,t-1}$ has an edge to $r_t$. In this example, we consider the case when the control signals are random, so there is no edge between $\mathbf{s}_t$ and $a_t$.

## 3 What, Where, and How to Adapt in RL

We first assume that the environment model in Eq. 1 is known (we will explain how to learn it in Sec. 3.1), and characterize which changes have an effect on the policy transfer to the target domain. In Eq. 1, we allow the model to change across domains, including all involved functions, and we leverage $\boldsymbol{\theta}_k$ to capture the changes in a compact way. The varying model implies that the optimal policy function may also vary across domains. How can we then characterize the changes in the optimal policy function in a compact way, as we did in the model? Interestingly, we find that the change factor $\boldsymbol{\theta}_k$ and the latent state $\mathbf{s}_t$ are sufficient for policy learning, but not every dimension of $\boldsymbol{\theta}_k$ or $\mathbf{s}_t$ is necessary, since they may not ever have an effect on the reward, even in future steps. We first give the definitions of *compact domain-shared representations* and *compact domain-specific representations*, according to the graph structure, and we further show that they are the minimal set of dimensions that suffice for policy learning across domains (proof in Appendix).

**Definition 1.** *Given the graphical representation of an environment model $\mathcal{G}$ that is encoded in the binary masks $\mathbf{c}^{\cdot \to \cdot}$, we define recursively the representations that affect the reward in the future as:*

- *compact domain-shared representations $\mathbf{s}_t^{min}$: the latent state components $s_{i,t} \in \mathbf{s}_t$ that either*
  - *have an edge to the reward in the next time-step $r_{t+1}$, i.e., $\mathbf{c}_i^{\mathbf{s} \to r} = 1$, or*
  - *have an edge to another state component in the next time-step $s_{j,t+1}$, i.e., $\mathbf{c}_{j,i}^{\mathbf{s} \to \mathbf{s}} = 1$, such that the same component at time $t$ is a compact domain-shared representation, i.e., $s_{j,t} \in \mathbf{s}_t^{min}$;*
- *compact domain-specific representations $\boldsymbol{\theta}_k^{min}$: the latent change factors $\theta_{i,k} \in \boldsymbol{\theta}_k$ that either:*
  - *have an edge to the reward in the next time-step $r_{t+1}$, i.e., $\theta_{i,k} = \theta_k^r$ and $c^{\theta_k \to r} = 1$, or*
  - *have an edge to a state component $s_{j,t} \in \mathbf{s}_t^{min}$, i.e., $\mathbf{c}_{j,i}^{\theta_k \to s} = 1$.*

**Proposition 1.** *Under the assumption that the graph $\mathcal{G}$ is Markov and faithful to the measured data, the union of compact domain-specific $\boldsymbol{\theta}_k^{min}$ and compact shared representations $\mathbf{s}_t^{min}$ are the minimal and sufficient dimensions for policy learning across domains.*

For the example in Fig. 1, $\mathbf{s}_t^{min} = (s_{1,t}, s_{d,t})$ and $\boldsymbol{\theta}_k^{min} = \{\theta_k^s, \theta_k^r\}$. Note that $\theta_k^o$ is never in $\boldsymbol{\theta}_k^{min}$, and thus if only the observation function changes, the optimal policy function $\pi_k^*$ remains the same across domains. For example in Cartpole a change of color does not affect the optimal policy. Moreover, if $c^{\theta_k \to r} = 1$, then $\theta_k^r \in \boldsymbol{\theta}_k^{min}$, which is the case for multi-task learning.

### 3.1 Simultaneous Estimation of Domain-Varying Models

In this section, we give the estimation procedure of the environment model in Eq. 1 from observed sequences $\{\{\langle \mathbf{y}_{t,k}, a_{t,k} \rangle\}_{t=1}^T\}_{k=1}^n$ from each source domain $k$, where $\mathbf{y}_{t,k} = (o_{t,k}^\top, r_{t,k}^\top)^\top$ are the observations and reward at time $t$ in domain $k$. Instead of estimating the model in each domain

separately, we estimate models from different domains simultaneously, by exploiting commonalities across domains while at the same time preserving specific information for each domain. In particular, we propose the Multi-model Structured Sequential Variational Auto-Encoder (**MiSS-VAE**), which contains the following three essential components. (1) *"Sequential VAE" component* handles the sequential data, with the underlying latent states satisfying an MDP. It is implemented by adding an LSTM (Hochreiter and Schmidhuber, 1997) to encode the sequential information in the encoder to learn the inference model $q_\phi(\mathbf{s}_{t,k}|\mathbf{s}_{t-1,k}, \mathbf{y}_{1:t,k}, a_{1:t-1,k}; \boldsymbol{\theta}_k)$. (2) *"Multi-model" component* handles models from different domains at the same time, using the domain index $k$ as an input and learning the domain-specific factors $\boldsymbol{\theta}_k$. (3) *"Structured" component*: exploits the structural information that is explicitly encoded with the binary masks, i.e., $\mathbf{c}^{\cdot \to \cdot}$ in Eq. 1. Here, the joint distribution of latent states are factorized according to structures, instead of being marginally independent as in traditional VAEs (Kingma and Welling, 2014). Fig. A3 in Appendix gives the diagram of neural network architecture in model training. Let $\mathbf{y}_{1:T}^{1:n} = \{\{\mathbf{y}_{t,k}\}_{t=1}^T\}_{k=1}^n$. By taking into account the above three components, we maximize the following objective function $\mathcal{L}$:

$$\mathcal{L}(\mathbf{y}_{1:T}^{1:n}; (\beta_1, \beta_2, \phi, \gamma, \mathbf{c}^{\cdot})) = \mathcal{L}^{\text{rec}}(\mathbf{y}_{1:T}^{1:n}; (\beta_1, \phi, \mathbf{c}^{\cdot})) + \mathcal{L}^{\text{pred}}(\mathbf{y}_{1:T}^{1:n}; (\beta_2, \phi)) - \mathcal{L}^{\text{KL}}(\mathbf{y}_{1:T}^{1:n}; (\phi, \gamma, \mathbf{c}^{\cdot})) - \mathcal{L}^{\text{reg}}.$$

In particular, $\mathcal{L}^{\text{rec}}$ is the reconstruction loss for both observed images and rewards, to learn the observation and the reward function, respectively. We also consider the one-step prediction loss $\mathcal{L}^{\text{pred}}$.

$$\mathcal{L}^{\text{rec}} = \sum_{k=1}^n \sum_{t=1}^{T-2} \mathbb{E}_{\mathbf{s}_{t,k} \sim q_\phi(\cdot|\boldsymbol{\theta}_k)} \{\log p_{\beta_1}(o_{t,k}|\mathbf{s}_{t,k}; \theta_k^o, c^{\theta_k \to o}, \mathbf{c}^{\mathbf{s} \to o}) + \log p_{\beta_1}(r_{t+1,k}|\mathbf{s}_{t,k}, a_{t,k}; \theta_k^r, c^{\theta_k \to r}, \mathbf{c}^{\mathbf{s} \to r}, c^{a \to r})\},$$

$$\mathcal{L}^{\text{pred}} = \sum_{k=1}^n \sum_{t=1}^{T-2} \mathbb{E}_{\mathbf{s}_{t,k} \sim q_\phi(\cdot|\boldsymbol{\theta}_k)} \{\log p_{\beta_2}(o_{t+1,k}|\mathbf{s}_{t,k}, \theta_k^o, \theta_k^s) + \log p_{\beta_2}(r_{t+2,k}|\mathbf{s}_{t,k}, a_{t+1,k}; \theta_k^r, \theta_k^s)\},$$

where $p_{\beta_1}$ and $p_{\beta_2}$ denote the generative models with parameters $\beta_1$ and $\beta_2$, respectively, that are shared across domains, and $q_\phi$ the inference model with shared parameters $\phi$. We also use the following KL-divergence loss to constrain the latent space:

$$\mathcal{L}^{\text{KL}} = \lambda_0 \sum_{k=1}^n \sum_{t=2}^T \text{KL}\big(q_\phi(\mathbf{s}_{t,k}|\mathbf{s}_{t-1,k}, \mathbf{y}_{1:t,k}, a_{1:t-1,k}; \boldsymbol{\theta}_k) \| p_\gamma(\mathbf{s}_{t,k}|\mathbf{s}_{t-1,k}, a_{t-1,k}; \theta_k^s, \mathbf{C}^{\mathbf{s} \to \mathbf{s}}, \mathbf{c}^{a \to \mathbf{s}}, \mathbf{C}^{\theta_k \to \mathbf{s}})\big),$$

where we explicitly model the transition dynamics $p_\gamma$ with the parameters $\gamma$ shared across domains; this is essential for establishing a Markov chain in latent space and learning a representation for long-term predictions. Moreover, the KL loss helps to constrain the latent space to (1) ensure that the disentanglement between the inferred latent factors $q(s_{i,t}|\cdot)$ and $q(s_{j,t}|\cdot)$ for $i \neq j$, since we do not consider the instantaneous connections among state dimensions, and (2) ensure that the latent representations $\mathbf{s}_t$ are maximally compressive about the observed high-dimensional data. Furthermore, according to the edge-minimality property (Zhang and Spirtes, 2011) and the minimal change principle (Ghassami et al., 2018), we add sparsity constraints on structural matrices and on the change of domain-specific factors across domains, respectively, to achieve better identifiability:

$$\mathcal{L}^{\text{reg}} = \lambda_1 \|\mathbf{c}^{\mathbf{s} \to o}\|_1 + \lambda_2 \|\mathbf{c}^{\mathbf{s} \to r}\|_1 + \lambda_3 \|c^{a \to r}\|_1 + \lambda_4 \|\mathbf{C}^{\mathbf{s} \to \mathbf{s}}\|_1 + \lambda_5 \|\mathbf{c}^{a \to \mathbf{s}}\|_1 + \lambda_6 \|\mathbf{C}^{\theta_k \to \mathbf{s}}\|_1 + \lambda_7 \sum_{1 \leq j,k \leq n} |\boldsymbol{\theta}_j - \boldsymbol{\theta}_k|.$$

Note that besides the shared parameters $\{\beta_1, \beta_2, \phi, \gamma\}$, the structural relationships (encoded in binary masks $\mathbf{c}$) are also involved in the shared parameters. Each factor in $p_\phi$, $p_{\beta_i}$, and $p_\gamma$ is modeled with a mixture of Gaussians, because with a suitable number of Gaussians, it can approximate a wide class of continuous distributions. Moreover, in model estimation, the domain-specific factors $\boldsymbol{\theta}_k = \{\theta_k^o, \theta_k^s, \theta_k^r\}$ are treated as parameters; they are constant within the same domain, but may differ in different domains. We explicitly consider $\boldsymbol{\theta}_k$ not only in the generative models $p_{\beta_i}$ and $p_\gamma$, but also in the inference model $q_\phi$. In this way, except for $\boldsymbol{\theta}_k$, all other parameters in MiSS-VAE are shared across domains, so that all we need to update in the target domain is the low-dimensional $\boldsymbol{\theta}_k$, which greatly improves the sample efficiency and the statistical efficiency in the target domain.

### 3.2 LOW-COST AND INTERPRETABLE POLICY TRANSFER

After identifying what and where to transfer, we show how to adapt. Instead of learning the optimal policy in each domain separately, which is time and sample inefficient, we leverage a multi-task learning strategy: policies in different domains are optimized at the same time exploiting both commonalities and differences across domains. Given the compact domain-shared $\mathbf{s}_t^{min}$ and domain-specific representations $\boldsymbol{\theta}_k^{min}$, we represent the optimal policies across domains in a unified way:

$$a_t = \pi^*(\mathbf{s}_t^{min}, \boldsymbol{\theta}_k^{min}), \tag{2}$$

where $\boldsymbol{\theta}_k^{min}$ explicitly and compactly encodes the changes in the policy function in each domain $k$, and all other parameters in the optimal policy function $\pi^*$ are shared across domains. In other

words, by learning $\pi^*$ in the source domains, and estimating the value of the change factor $\boldsymbol{\theta}_{\text{target}}^{min}$ and inferring latent states $\mathbf{s}_{\text{target}}^{min}$ from the target domain, we can immediately derive the optimal policy in the target domain without further policy optimization by just applying Eq. 2.

The AdaRL framework answers what and where the change factors are and which change factors need to adapt across domains in an interpretable way. Moreover, AdaRL only requires a few samples to update the low-dimensional domain-specific parameters $\boldsymbol{\theta}_{\text{target}}^{min}$ to achieve the optimal policy in the target domain, without further policy optimization, achieving the low cost. We provide the pseudocode for the AdaRL algorithm in Alg. 1. The algorithm has three parts: (1) data collection with a random policy or any initial policy from $n$ source domains (line 2), (2) model estimation from the $n$ source domains with multi-task learning (lines 2-3, see Sec. 3.1 for details), and (3) learning the optimal policy $\pi^*$ with deep Q-learning, by making use of domain-specific factors and the inferred domain-shared state representations (lines 4-21). Specifically, because we do not directly observe the states $\mathbf{s}_t$, we infer $q(\mathbf{s}_{t+1,k}^{min}|o_{\leq t+1,k}, r_{\leq t+1,k}, a_{\leq t,k}, \boldsymbol{\theta}_k^{min})$ and sample $\mathbf{s}_{t+1,k}^{min}$ from its posterior, for the $k$th domain (lines 7 and 13). Moreover, the action-value function $Q$ is learned by considering the averaged error over the $n$ source domains (line 18). AdaRL can be implemented with a wide class of policy-learning algorithms, e.g., DDPG (Lillicrap et al., 2015), Q-learning (Mnih et al., 2015), and Actor-Critic methods (Schulman et al., 2016; Mnih et al., 2016). Then, in the target domain, we only need to collect a few rollouts to estimate the low-dimensional domain-specific representations $\boldsymbol{\theta}_{\text{target}}^{min}$, with all other parameters being fixed (lines 22-23).

---

**Algorithm 1** (AdaRL with Domains Shifts)

---

1: Initialize action-value function $Q$, target action-value function $Q'$, and replay buffer $\mathcal{B}$.
2: Record multiple rollouts for each source domain $k(k = 1, \cdots, n)$ and estimate the model in Eq.1.
3: Identify the dimension indices of $\mathbf{s}_t^{min}$ and the values of $\boldsymbol{\theta}_k^{min}$ according to the learned model.
4: **for** episode = 1, ..., M **do**
5:     **for** source domain k = 1, ..., n **do**
6:         Receive initial observations $o_{1,k}$ and $r_{1,k}$ for the $k$-th domain.
7:         Infer the posterior $q(\mathbf{s}_{1,k}^{min}|o_{1,k}, r_{1,k}, \boldsymbol{\theta}_k^{min})$ and sample initial inferred state $\mathbf{s}_{1,k}^{min}$.
8:     **end for**
9:     **for** timestep t = 1, ..., T **do**
10:        **for** source domain k = 1, ..., n **do**
11:            Select $a_{t,k}$ randomly with probability $\epsilon$; otherwise $a_{t,k} = \arg\max_a Q(\mathbf{s}_{t,k}^{min}, a, \boldsymbol{\theta}_k^{min})$.
12:            Execute action $a_{t,k}$, and receive reward $r_{t+1,k}$ and observation $o_{t+1,k}$ in the $k$th domain.
13:            Infer the posterior $q(\mathbf{s}_{t+1,k}^{min}|o_{\leq t+1,k}, r_{\leq t+1,k}, a_{\leq t,k}, \boldsymbol{\theta}_k^{min})$ and sample $\mathbf{s}_{t+1,k}^{min}$.
14:            Store transition $(\mathbf{s}_{t,k}^{min}, a_{t,k}, r_{t+1,k}, \mathbf{s}_{t+1,k}^{min}, \boldsymbol{\theta}_k^{min})$ in reply buffer $\mathcal{B}$.
15:        **end for**
16:        Randomly sample a minibatch of $N$ transitions $(\mathbf{s}_{i,j}^{min}, a_{i,j}, r_{i+1,j}, \mathbf{s}_{i+1,j}^{min}, \boldsymbol{\theta}_j^{min})$ from $\mathcal{B}$.
17:        Set $y_{i,j} = r_{i+1,j} + \lambda \max_{a'} Q'(s_{i+1,j}^{min}, a', \boldsymbol{\theta}_j^{min})$.
18:        Update action-value function $Q$ by minimizing the loss:

$$L = \frac{1}{n * N} \sum_{i,j} (y_{i,j} - Q(s_{i,j}^{min}, a_{i,j}, \boldsymbol{\theta}_j^{min}))^2.$$

19:     **end for**
20:     Update the target network $Q'$: $Q' = Q$.
21: **end for**
22: Record a few rollouts from the target domain.
23: Estimate the values of $\boldsymbol{\theta}_{\text{target}}^{min}$ for the target domain, with all other parameters fixed.

---

## 3.3 THEORETICAL PROPERTIES

Below we show the conditions under which we can identify the true graph $\mathcal{G}$ from observational data, even when the model in Eq. 1 is unknown. Furthermore, we derive a generalization bound of the state-value function under the PAC-Bayes framework (McAllester, 1999).

**Theorem 1** (Structural Identifiability). *Suppose the underlying states $\mathbf{s}_t$ are observed, i.e., Eq. (1) is an MDP. Then under the Markov condition and faithfulness assumption, the structural matrices $\mathbf{C}^{\mathbf{s} \rightarrow \mathbf{s}}, \mathbf{c}^{a \rightarrow \mathbf{s}}, \mathbf{c}^{\mathbf{s} \rightarrow r}, c^{a \rightarrow r}, \mathbf{C}^{\theta_k \rightarrow \mathbf{s}}, and c^{\theta_k \rightarrow r} are identifiable.*

This theorem shows that in the MDP scenario, where the underlying states are observed and $c^{\theta_k \to o}$ and $\mathbf{c}^{\mathbf{s} \to o}$ are not considered by definition, we can uniquely determine the structural relationships over $\{\mathbf{s}_{t-1}, \mathbf{s}_t, a_{t-1}, r_t, \boldsymbol{\theta}_k\}$, i.e., the Dynamic Bayesian network $\mathcal{G}$, from observed data under mild conditions, without knowing the generative environment model. Even if $\boldsymbol{\theta}_k$ is not directly observed, we can identify which state dimension changes and if there is a change in the reward function.

Suppose there are $n$ source domains, and for the $k$th domain, we have $S_k = \big( (\mathbf{s}_{1,k}, v^*(\mathbf{s}_{1,k})), \cdots, (\mathbf{s}_{m_k,k}, v^*(\mathbf{s}_{m_k,k})) \big)$, where $m_k$ is the number of samples from the $k$th domain, $\mathbf{s}_{\cdot,k}$ is a state sampled from the $k$th domain, and $v^*(\mathbf{s}_{\cdot,k})$ is its corresponding optimal state-value. For any value function $h_{\boldsymbol{\theta}_k^{min}}(\cdot)$ parameterized by $\boldsymbol{\theta}_k^{min}$, we define the loss function $\ell(h_{\boldsymbol{\theta}_k^{min}}, (\mathbf{s}_{k,i}, v^*(\mathbf{s}_{i,k}))) = D_{dist}(h_{\boldsymbol{\theta}_k^{min}}(\mathbf{s}_{i,k}), v^*(\mathbf{s}_{i,k}))$, where $D_{dist}$ is a distance function that measures the discrepancy between the learned value and the optimal value. The following theorem gives a generalization bound of the state-value function under the PAC-Bayes framework.

**Theorem 2** (Generalization Bound). *Let $\mathcal{Q}$ be an arbitrary distribution over $\boldsymbol{\theta}_k^{min}$ and $\mathcal{P}$ the prior distribution over $\boldsymbol{\theta}_k^{min}$. Then for any $\delta \in (0,1]$, with probability at least $1-\delta$, the following inequality holds uniformly for all $\mathcal{Q}$,*

$$er(\mathcal{Q}) \leq \frac{1}{n} \sum_{k=1}^{n} \left\{ \hat{er}(\mathcal{Q}, S_k) + \sqrt{\frac{1}{2(m_k-1)} \left( D_{KL}(\mathcal{Q}||\mathcal{P}) + \log \frac{2nm_k}{\delta} \right)} + \sqrt{\frac{1}{2(n-1)} \left( D_{KL}(\mathcal{Q}||\mathcal{P}) + \log \frac{2n}{\delta} \right)} \right\},$$

*where $er(\mathcal{Q})$ and $\hat{er}(\mathcal{Q}, S_k)$ are the generalization error and the training error between the estimated value and the optimal value, respectively.*

Theorem 2 states that with high probability the generalization error $er(\mathcal{Q})$ is upper bounded by the empirical error plus two complexity terms. Specifically, the first one is the average of the task-complexity terms from the observed domains, which converges to zero in the limit of samples in each domain, i.e., $m_k \to \infty$. The second is an environment-complexity term, which converges to zero if infinite domains are observed, i.e., $n \to \infty$. Moreover, if assuming different dimensions of $\boldsymbol{\theta}_k^{min}$ are independent, then $D_{KL}(\mathcal{Q}||\mathcal{P}) = \sum_{i=1}^{|\boldsymbol{\theta}_k^{min}|} D_{KL}(\mathcal{Q}_i||\mathcal{P}_i)$, which indicates that a low-dimensional $\boldsymbol{\theta}_k^{min}$ usually has a smaller KL divergence, so does the upper bound of the generalization error.

## 4 EVALUATION

We modify the Cartpole and Atari Pong environments in OpenAI Gym (Brockman et al., 2016). Here, we present a subset of the results; see Appendix for the complete results and the detailed settings. We consider changes in the state dynamics (e.g., the change of gravity or cart mass in Cartpole, change of orientation in Pong), changes in observations (e.g., different noise levels in images or different colors in Pong), and changes in reward functions (e.g., different reward functions in Pong based on the contact point of the ball), as shown in Fig. 2 for Pong. For each of these factors, we take into account both *interpolation* (where the factor value in the target domain is in the support of that in source domains), and *extrapolation* (where it is out of the support w.r.t. the source domains). We train on $n$ source domains based on the trajectory data generated by a random policy. In Cartpole, for each domain we collect 10000 trials with 40 steps. For Pong experiments, each domain contains 40 episodes data and each of them takes a maximum of 10000 steps. In the target domain we consider different sample sizes with $N_{\text{target}} = \{20, 50, 10000\}$ to estimate $\boldsymbol{\theta}_{\text{target}}^{min}$. For both games, we evaluate the POMDP case, where the inputs are high-dimensional images; note that we did not stack multiple frames, so some properties (e.g., velocity) are not observed, resulting in a POMDP. For Cartpole, we also consider the MDP case, where the true states (cart position and velocity, pole angle and angular velocity) are used as the input to the model. In Cartpole, we also experiment with multiple factors changing at the same time (e.g., gravity and mass change concurrently in the target domain).

**Modified Cartpole setting** The Cartpole problem consists of a cart and a vertical pendulum attached to the cart using a passive pivot joint. The task is to prevent the vertical pendulum from falling by putting a force on the cart to move it left or right. We introduce two change factors for the state dynamics $\theta_k^s$: varying gravity and varying mass of the cart. In terms of changes on the observation function $\theta_k^o$, we add Gaussian noise on the images. Since $\theta_k^o$ does not influence the optimal policy (as shown in Prop. 1), we need it only for the model estimation, but not for policy optimization. Moreover, if $\boldsymbol{\theta}_k = \{\theta_k^o\}$, the optimal policy is shared across domains.

Figure 2: Illustrations of the change factors on modified Pong game.

|  | Oracle Upper bound | Non-t lower bound | CAVIA (Zintgraf et al., 2019) | PEARL (Rakelly et al., 2019) | AdaRL* Ours w/o masks | AdaRL Ours |
|---|---|---|---|---|---|---|
| G_in | 2486.1 (±369.7) | 1098.5 ● (±472.1) | 1603.0 (±877.4) | 1647.4 (±617.2) | 1940.5 (±841.7) | 2217.6 (±981.5) |
| G_out | 693.9 (±100.6) | 204.6 ● (±39.8) | 392.0 ● (±125.8) | 434.5 ● (±102.4) | 439.5 ● (±157.8) | **508.3** (±138.2) |
| M_in | 2678.2 (±630.5) | 748.5 ● (±342.8) | 2139.7 (±859.6) | 1784.0 (±845.3) | 1946.2 ● (±496.5) | 2260.2 (±682.8) |
| M_out | 1405.6 (±368.0) | 371.0 ● (±92.5) | 972.6 ● (±401.4) | 793.9 ● (±394.2) | 874.5 ● (±290.8) | **1001.7** (±273.3) |
| G_in & M_in | 1984.2 (±871.3) | 365.0 ● (±144.5) | 1012.5 ● (±664.9) | 1260.8 ● (±792.0) | 1157.4 ● (±578.5) | **1428.4** (±495.6) |
| G_out & M_out | 939.4 (±270.5) | 336.9 ● (±139.6) | 648.2 ● (±481.5) | 544.32 ● (±175.2) | 596.0 ● (±184.3) | **689.4** (±272.5) |

Table 1: Average final scores on modified Cartpole (MDP) with $N_{target} = 50$. The best non-oracle results w.r.t. the mean are marked in red, while **bold** indicates a statistically significant result w.r.t. all the baselines, and "●" indicates the baseline for which the improvements of AdaRL are statistically significant (via Wilcoxon signed-rank test at $5\%$ significance level). G and M denote the gravity and mass respectively, and "*in" and "*out" denote the interpolation and extrapolation, respectively.

**Modified Pong setting** In Pong, one of the established Atari benchmarks (Bellemare et al., 2013), the agent controls a paddle moving up and down vertically, aiming at hitting the ball. We consider changes in observation function $\theta_k^o$, state dynamics $\theta_k^s$, and reward function $\theta_k^r$, as shown in Fig. 2. We consider three change factors on perceived signals $\theta_k^o$: different image sizes, different image colors, and different noise levels. For the setting with different image colors, we use RGB images as inputs and consider source domains with varying RGB colors {original, green, red} and target domains with colors {yellow, white}, but for other settings, we convert the images to grayscale as input. To change the state dynamics, we rotate the images $\omega$ degrees clockwise. To test the changes in the reward function, we modify the reward as a function of the distance between contact point and the central point of the paddle, denoted by $d$, as opposed to the original Pong in which it is constant ($-1$ or $+1$) when the agent or the opponent misses the pong. We denote by $L$ the half-length of the paddle and formulate two groups of reward functions: (1) Linear reward: $r_t = \frac{\alpha d}{L}$; and (2) Non-linear reward: $r_t = \frac{\alpha L}{d+3L}$, where $\alpha$ varies across domains.

**Baselines** In the MDP setting, we compare AdaRL with CAVIA (Zintgraf et al., 2019) and PEARL (Rakelly et al., 2019). In the POMDP setting, we compare with PNN (Rusu et al., 2016), PSM (Agarwal et al., 2021) and MQL (Fakoor et al., 2020). We also compare with *AdaRL\**, a version of AdaRL that does not learn the binary masks $c^{\cdot \rightarrow \cdot}$ and therefore does not use any structural information. All of these methods use the same number of samples $N_{target}$ from the target domain. We also compare with: 1) *Non-t*, a vanilla non-transfer baseline that pools data from all source domains and learns a fixed model; and 2) an oracle baseline, which is **completely** trained on the target domain with model-free exploration. For a fair comparison, we use the same policy learning algorithm, Double DQN (Van Hasselt et al., 2016), for all methods. As opposed to MAML and PNN, AdaRL only uses the $N_{target}$ samples to estimate $\theta_{target}^{min}$, without any policy optimization.

**Results** We measure performance by the mean and standard deviation of the final scores over 30 trials with different random seeds, as well as testing the significance with the Wilcoxon signed-rank test (Conover, 1999). As shown in Tables 1, 2 and 3 [2], AdaRL consistently outperforms the baselines across most change factors in the MDP and POMDP case for modified Cartpole, and

---

[2]In Table 1-3, "●" indicates the baselines for which the improvements of AdaRL are statistically significant (via Wilcoxon signed-rank test at $5\%$ significance level).

| | Oracle
Upper bound | Non-t
lower bound | PNN
(Rusu et al., 2016) | PSM
(Agarwal et al., 2021) | MTQ
(Fakoor et al., 2020) | AdaRL*
Ours w/o masks | AdaRL
Ours |
|---|---|---|---|---|---|---|---|
| G_in | 1930.5
($\pm$1042.6) | 1031.5 ●
($\pm$837.9) | 1268.5 ●
($\pm$699.0) | 1439.8 ●
($\pm$427.6) | 1517.9
($\pm$883.6) | 1460.6
($\pm$497.5) | 1697.4
($\pm$1002.3) |
| G_out | 408.6
($\pm$67.2) | 69.7 ●
($\pm$19.4) | 307.9 ●
($\pm$100.4) | 273.8 ●
($\pm$92.6) | 330.6
($\pm$109.8) | 298.6 ●
($\pm$69.3) | 353.4
($\pm$79.6) |
| M_in | 2004.9
($\pm$404.3) | 608.5 ●
($\pm$222.8) | 1600.8 ●
($\pm$463.5) | 1891.5
($\pm$638.4) | 1735.6 ●
($\pm$398.7) | 1884.5
($\pm$429.7) | 1912.8
($\pm$378.5) |
| M_out | 1498.6
($\pm$625.4) | 216.4 ●
($\pm$77.3) | 987.6 ●
($\pm$368.5) | 1032.7 ●
($\pm$634.0) | 862.2 ●
($\pm$300.4) | 1219.5
($\pm$1014.3) | 1467.5
($\pm$837.2) |
| N_in | 8640.5
($\pm$3086.1) | 942.0 ●
($\pm$207.5) | 3952.4 ●
($\pm$1024.9) | 5279.6 ●
($\pm$1969.7) | 6927.3 ●
($\pm$2464.8) | 5540.8 ●
($\pm$2013.6) | **7817.4**
($\pm$3009.5) |
| N_out | 4465.2
($\pm$667.3) | 1002.8 ●
($\pm$335.2) | 1137.1 ●
($\pm$384.6) | 2740.9 ●
($\pm$511.5) | 3298.5 ●
($\pm$537.8) | 2018.9 ●
($\pm$685.4) | **3640.9**
($\pm$841.0) |

Table 2: Average final scores on modified Cartpole (POMDP) with $N_{target} = 50$. The best non-oracle results are marked in **red**. G, M, and N denote the gravity, mass, and noise respectively.

| | Oracle
Upper bound | Non-t
lower bound | PNN
(Rusu et al., 2016) | PSM
(Agarwal et al., 2021) | MTQ
(Fakoor et al., 2020) | AdaRL*
Ours w/o masks | AdaRL
Ours |
|---|---|---|---|---|---|---|---|
| O_in | 18.65
($\pm$2.43) | 6.18 ●
($\pm$2.43) | 9.70 ●
($\pm$2.09) | 11.61 ●
($\pm$3.85) | 15.79 ●
($\pm$3.26) | 14.27 ●
($\pm$1.93) | **18.97**
($\pm$2.00) |
| O_out | 19.86
($\pm$1.09) | 6.40 ●
($\pm$3.17) | 9.54 ●
($\pm$2.78) | 10.82 ●
($\pm$3.29) | 10.82 ●
($\pm$4.13) | 12.67 ●
($\pm$2.49) | **15.75**
($\pm$3.80) |
| C_in | 19.35
($\pm$0.45) | 8.53 ●
($\pm$2.08) | 14.44 ●
($\pm$2.37) | 19.02
($\pm$1.17) | 16.97 ●
($\pm$2.02) | 18.52 ●
($\pm$1.41) | 19.14
($\pm$1.05) |
| C_out | 19.78
($\pm$0.25) | 8.26 ●
($\pm$3.45) | 14.84 ●
($\pm$1.98) | 17.66 ●
($\pm$2.46) | 15.45 ●
($\pm$3.30) | 17.92
($\pm$1.83) | 19.03
($\pm$0.97) |
| S_in | 18.32
($\pm$1.18) | 6.91 ●
($\pm$2.02) | 11.80 ●
($\pm$3.25) | 12.65 ●
($\pm$3.72) | 13.68 ●
($\pm$3.49) | 14.23 ●
($\pm$3.19) | **16.65**
($\pm$1.72) |
| S_out | 19.01
($\pm$1.04) | 6.60 ●
($\pm$3.11) | 9.07 ●
($\pm$4.58) | 8.45 ●
($\pm$4.51) | 11.45 ●
($\pm$2.46) | 12.80 ●
($\pm$2.62) | **17.82**
($\pm$2.35) |
| N_in | 18.48
($\pm$1.25) | 5.51 ●
($\pm$3.88) | 12.73 ●
($\pm$3.67) | 11.30 ●
($\pm$2.58) | 12.67 ●
($\pm$3.84) | 13.78 ●
($\pm$2.15) | **16.84**
($\pm$3.13) |
| N_out | 18.26
($\pm$1.11) | 6.02 ●
($\pm$3.19) | 13.24 ●
($\pm$2.55) | 11.26 ●
($\pm$3.15) | 15.77 ●
($\pm$2.12) | 14.65 ●
($\pm$3.01) | **18.30**
($\pm$2.24) |

Table 3: Average final scores on modified Pong (POMDP) with $N_{target} = 50$. The best non-oracle are marked in **red**. O, C, S, and N denote the orientation, color, size, and noise factors, respectively.

in the POMDP case for Pong for $N_{target} = 50$. As ablation studies, to see the effect of learning the graphical structure, we also compare with $AdaRL*$, which does not learn the binary masks $c$, but just assumes everything is fully connected. Learning the graphical structure improves the performances significantly, and without it $AdaRL*$ is generally comparable to baselines. We provide results with $N_{target} = \{20, 50, 10000\}$ in Appendix, showing that the performance gains are larger at smaller sample sizes. Furthermore, we consider the change of reward functions (see Table A12-14 in Appendix). More detailed experimental results are provided in Appendix, including the average score across different $N_{target}$, policy learning curves and an analysis of the estimated $\theta_k$ w.r.t. real change factor. Interestingly, in the Cartpole case, the estimated $\theta_k$ matches the physical quantities that are being changed across the domains. In particular, the estimated $\theta_k$ for gravity and noise are linear mappings of the gravity and noise level values. For the mass-varying case, the learned $\boldsymbol{\theta}_k^s$ is a nonlinear monotonic function of the mass, which matches the influence of the mass on the dynamics.

## 5 CONCLUSIONS AND FUTURE WORK

In this paper, we proposed AdaRL, a principled framework for transfer RL. AdaRL learns a latent representation with domain-shared and domain-specific components across source domains, uses it to learn an optimal policy parameterized by the domain-specific parameters, and applies it to a new target domain. It is achieved without any further policy optimization, but just by estimating the values of the domain-specific parameters in the target domain, which can be accomplished with a few target-domain data. As opposed to previous work, AdaRL can model changes in the state dynamics, observation function and reward function in an unified manner, and exploit the factorization to improve the data efficiency and adapt faster with fewer samples. Further directions include exploiting the target domain to fine-tune the policy and handling the out-of-distribution data. Moreover, exploring an alternative to the reconstruction loss, e.g., using the contrastive loss (Laskin et al., 2020), might also improve the training efficiency. Finally, an exciting next step is to transfer knowledge across different tasks, e.g., different Atari games.

## REPRODUCIBILITY STATEMENT

Proofs of all our theoretical results are given in Appendix A1, A2, and A3 with disclosure of all assumptions. More details about the model estimation are given in Appendix A4. Appendix A5 provides the complete experimental details and results. All datasets used are publicly available or instructions are provided on how to generate them in Appendix 5 and 6. A description of the hyperparameters and network architectures used is included in Appendix 6. Source code is given at https://github.com/Adaptive-RL/AdaRL-code, providing also a complete description of our experimental environment, configuration files and instructions on the reproduction of our experiments.

## ACKNOWLEDGEMENT

KZ would like to acknowledge the support by the National Institutes of Health (NIH) under Contract R01HL159805, by the NSF-Convergence Accelerator Track-D award #2134901, by the United States Air Force under Contract No. FA8650-17-C7715, and by a grant from Apple. BH would like to acknowledge the support by Apple PhD fellowship in AI/ML. Part of the work was performed while BH was interning at the MIT-IBM Watson AI Lab.

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

# APPENDIX FOR "ADARL: WHAT, WHERE, AND HOW TO ADAPT IN TRANSFER REINFORCEMENT LEARNING"

Appendix organization:

## A1 PROOF OF PROPOSITION 1

We first review the definitions of d-separation, the Markov condition, and the faithfulness assumption (Spirtes et al., 1993; Pearl, 2000), which will be used in the proof.

Given a directed acyclic graph $G = (\mathbf{V}, \mathbf{E})$, where $\mathbf{V}$ is the set of nodes and $\mathbf{E}$ is the set of directed edges, we can define a graphical criterion that expresses a set of conditions on the paths.

**Definition A1** (d-separation (Pearl, 2000)). *A path $p$ is said to be d-separated by a set of nodes $\mathbf{Z} \subseteq \mathbf{V}$ if and only if (1) $p$ contains a chain $i \to m \to j$ or a fork $i \leftarrow m \to j$ such that the middle node $m$ is in $Z$, or (2) $p$ contains a collider $i \to m \leftarrow j$ such that the middle node $m$ is not in $\mathbf{Z}$ and such that no descendant of $m$ is in $\mathbf{Z}$.*

*Let X, Y, and Z be disjunct sets of nodes. Z is said to d-separate X from Y (denoted as $X \perp_d Y|Z$) if and only if Z blocks every path from a node in X to a node in Y.*

**Definition A2** (Global Markov Condition (Spirtes et al., 1993; Pearl, 2000)). *A distribution $P$ over $\mathbf{V}$ satisfies the global Markov condition on graph $G$ if for any partition $(X, Z, Y)$ such that $X \perp_d Y|Z$*

$$P(X, Y|Z) = P(X|Z)P(Y|Z).$$

*In other words, X is conditionally independent of Y given Z, which we denote as $X \perp\!\!\!\perp Y|Z$.*

**Definition A3** (Faithfulness Assumption (Spirtes et al., 1993; Pearl, 2000)). *There are no independencies between variables that are not entailed by the Markov Condition.*

If we assume both of these assumptions, then we can use d-separation as a criterion to read all of the conditional independences from a given DAG $G$. In particular, for any disjoint subset of nodes $\mathbf{X,Y,Z} \subseteq \mathbf{V}$: $\mathbf{X} \perp\!\!\!\perp \mathbf{Y}|\mathbf{Z} \iff \mathbf{X} \perp_d \mathbf{Y}|\mathbf{Z}$.

In our case we can represent the generative model in Eq. 1 as a Dynamic Bayesian Network(DBN) $\mathcal{G}$ (Murphy, 2002) over the variables $\{\mathbf{s}_{t-1}, a_{t-1}, o_{t-1}, r_t, \mathbf{s}_t, \boldsymbol{\theta}_k\}$, where the binary masks $c^{\rightarrow\cdot}$ represent edges or sets of edges, as shown in Fig. A1. As is typical in DBN we assume that the graph is invariant across different timesteps. We assume $\boldsymbol{\theta}_k$ are constant across the different timesteps. We add to the image also the cumulative reward R. In practice we will focus instead on the cumulative *future* reward $R_{t+1} = \sum_{\tau=t+1}^{T} \gamma^{\tau-t-1} r_\tau$, which only considers the contributions of the $r_\tau$ in the future with respect to the current timestep $t$.

In order to prove Proposition 1, we first need to prove that the compact shared representations $\mathbf{s}_t^{min}$ and compact shared representations $\boldsymbol{\theta}_k^{min}$ are all the state and change factors dimensions, respectively, that are conditionally independent of $a_t$ given the future cumulative reward $R_{t+1}$, even given all other variables:

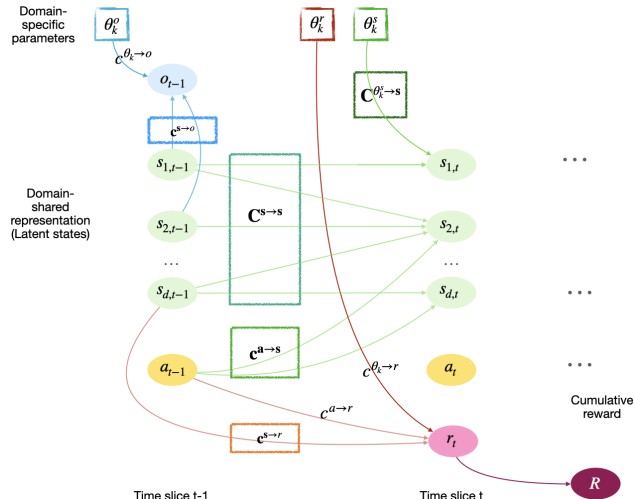

Figure A1: Graphical representation of the generative model in Eq. 1. In the top of the figure, the square boxes are the domain-specific parameters $\boldsymbol{\theta}_k$, which are constant in time, while the rectangular boxes represent the binary masks that encode the edges. R represents the cumulative reward

**Lemma A1.** *Under the assumption that the graph $\mathcal{G}$ is Markov and faithful to the measured data, a state dimension $s_{i,t} \in \mathbf{s}_t$ is part of $\mathbf{s}_t^{min}$ iff:*

$$s_{i,t} \not\perp\!\!\!\perp a_t | R_{t+1}, \tilde{s}_t \ \ \forall \tilde{s}_t \subseteq \{\mathbf{s}_t \setminus s_{i,t}\},$$

*Similarly, a change factor dimension $\theta_{i,k} \in \theta_k$ is part of $\boldsymbol{\theta}_k^{min}$ iff:*

$$\theta_{i,k} \not\perp\!\!\!\perp a_t | R_{t+1}, \tilde{l}_t, \ \ \forall \tilde{l}_t \subseteq \{\mathbf{s}_t, \{\boldsymbol{\theta}_k \setminus \theta_{i,k}\}\}.$$

*Proof.* We split the proof in two parts, the "only if" and the "if" part:

**"If conditionally dependent on $a_t$ given $R_{t+1}$ then in compact representation ":**

We first show that if $s_{i,t}$ satisfies the conditional dependence $s_{i,t} \not\perp\!\!\!\perp a_t | R_{t+1}, \tilde{s}_t, \ \ \forall \tilde{s}_t \subseteq \{\mathbf{s}_t \setminus s_{i,t}\}$, then it is part of $\mathbf{s}_t^{min}$, i.e. $s_{i,t}$ either has an edge to the reward in the next time-step $r_{t+1}$, or, recursively, it has an edge to another state component in the next time-step $s_{j,t+1}$, such that the same component at time step $t$, $s_{j,t+1} \in \mathbf{s}_t^{min}$. Note that this recursive definition collects all of the states $s_{i,t}$ that have an effect on future reward $r_{t+\tau}, \tau = \{1, \ldots, T - t\}$, either directly, or through the influence on other state components. Since all of these $r_{t+\tau}$ are influencing the cumulative future reward $R_{t+1}$, all of these components have an edge to $R_{t+1}$ as well. We prove it by contradiction. Suppose that $s_{i,t} \not\perp\!\!\!\perp a_t | R_{t+1}$ does not have a direct or indirect path to $r_{t+\tau}$, i.e. $s_{i,t} \in \mathbf{s}^{min}$. By assumption $a_t$ only affects future state $\mathbf{s}_{t+1}$, so $s_{i,t} \perp\!\!\!\perp a_t$. Then, according to the Markov and faithfulness conditions, $s_{i,t}$ is independent of $a_t$ conditioning on $R_{t+1}$, since there is no path that connects $s_{i,t} \to \cdots \to r_{t+\tau} \to R_{t+1} \leftarrow r_{t+1} \leftarrow a_t$ on which $R_{t+1}$ is a collider (i.e. a variable with two incoming edges), which is the only path which could introduce a conditional dependence. This contradicts the assumption.

Similarly we show that if $\forall \tilde{l}_t \subseteq \{\mathbf{s}_t, \{\boldsymbol{\theta}_k \setminus \theta_{i,k}\}\}$ the change factor dimension $\theta_{i,k} \not\perp\!\!\!\perp a_t | R_{t+1}, \tilde{l}_t$, then $\theta_{i,k} \in \boldsymbol{\theta}_k^{min}$, which similarly to previous case means it has a direct or indirect effect on $r_{t+\tau}$ and therefore $R_{t+1}$. By contradiction suppose that $\theta_{i,k} \not\perp\!\!\!\perp a_t | R_{t+1}, \tilde{l}_t$ for all previously defined $\tilde{l}_t$, but it is not a change parameter for the reward function $\theta_{i,k} \notin \theta_k^r$ with $c^{\theta_k \to r} = 1$, nor it is a change parameter for the state dynamics $\theta_{i,k} \in \theta_k^s$ with a direct or indirect path to $r_{t+\tau}$ for $\tau = 1, \ldots, T - t$. By assumption of our model, $\theta_{i,k}$ is never connected to $a_t$ directly, nor they might have a common cause, so $\theta_{i,k} \perp\!\!\!\perp a_t$. Then, according to the Markov condition, $\theta_{i,k}$ is independent of $a_t$ conditioning on $R_{t+1}$, which contradicts the assumption, since:

- if $\theta_{i,k} \notin \theta_k^r$ or $c^{\theta_k \to r} = 0$, then it means there is no path $\theta_{i,k} \to s_{i,t+\tau} \to r_{t+\tau} \to R_{t+1} \leftarrow r_{t+1} \leftarrow a_t$ for any $\tau \in \mathbb{N}$ that would be open by conditioning on $R_{t+1}$;

- if $\theta_{i,k} \in \theta_k^s$ but there is no directed path to $r_{t+\tau}$ for any $\tau \geq 1$, i.e. then there is also no directed path $\theta_{i,k} \rightarrow \cdots \rightarrow r_{t+\tau} \rightarrow R_{t+1} \leftarrow r_{t+1} \leftarrow a_t$ that would be open when we condition on $R_{t+1}$.

**"If in compact representation then conditionally dependent on $a_t$ given $R_{t+1}$":**

We next show that if $s_{i,t} \in \mathbf{s}_t^{min}$, which mean $s_{i,t}$ has a direct or indirect edge to $r_{t+\tau}$, $\tau = \{1, \ldots, T-t\}$, then $s_{i,t}$ satisfies the conditional dependence $s_{i,t} \not\perp\!\!\!\perp a_t | R_{t+1}, \tilde{s}_t, \ \forall \tilde{s}_t \subseteq \{\mathbf{s}_t \setminus s_{i,t}\}$. We prove it by contradiction. Suppose $s_{i,t}$ has a directed path to $r_{t+\tau}$ and $s_{i,t}$ is independent on $a_t$ given $R_{t+1}$ and a subset of other variables $\tilde{s}_t \subseteq \mathbf{s}_t \setminus s_{i,t}$. Since we assume that there are no instantaneous causal relations across the state dimensions, if $s_{i,t} \not\perp\!\!\!\perp a_t | R_{t+1}$ there can never be an $s_{j,t}$ such that $s_{i,t} \perp\!\!\!\perp a_t | R_{t+1}, s_{j,t}$. In this case, this means that $s_{i,t} \perp\!\!\!\perp a_t | R_{t+1}$ has to hold. Then according to the Markov and faithfulness assumptions, $s_{i,t}$ cannot have any directed path to any $r_{t+\tau} \forall \tau \geq 1$, because that any such path create a v-structure in the collider $R_{t+1}$, which would be open if we condition on $R_{t+1}$, contradicting the assumption.

Similarly, suppose $\theta_{i,k}$ is a dimension in $\boldsymbol{\theta}_k$ that has a directed path to $r_{t+\tau}$. We distinguish two cases, and show in neither can $\theta_{i,k}$ be independent of $a_t$ given $R_{t+1}$ and a subset of the other variables:

- if $\theta_{i,k} \in \theta_k^r$, then it cannot be independent of $a_t$ when we condition on $R_{t+1}$, which is a descendant of $r_{t+1}$ and therefore opens the collider path $\theta_k^r \leftarrow r_{t+1} \leftarrow a_t$;

- if $\theta_{i,k} \in \theta_k^s$, then at timestep $t$ it is always only connected to the corresponding $s_{i,t}$. So if there is a directed path $\pi$ to $r_{t+\tau}$, it has to go through $s_{i,t}$. While $\pi$ cannot be blocked by any subset of $\{\boldsymbol{\theta}_k \setminus \theta_{i,k}\}$, it can be blocked by conditioning on $s_{i,t}$, there are infinite future paths with the same structure, e.g. through $s_{i,t+1}$ that will not be blocked by conditioning only on variables at timestep $t$. Under the faithfulness and Markov assumption this means that $\theta_{i,k}$ cannot be independent from $a_t$ by conditioning on any subset of state variables at timestep $t$ or any other change parameters, which is a contradiction.

$\square$

We can now prove our main proposition:

**Proposition 1.** *Under the assumption that the graph $\mathcal{G}$ is Markov and faithful to the measured data, the union of compact domain-specific $\boldsymbol{\theta}_k^{min}$ and compact shared representations $\mathbf{s}_t^{min}$ are the minimal and sufficient dimensions for policy learning across domains.*

*Proof.* As shown in the previous lemma, in *compact domain-generalizable representations* $\mathbf{s}_t^{min}$ every dimension is dependent on $a_t$ given $R_{t+1}$ and any other variables, and every other dimension is independent of $a_t$ given $R_{t+1}$ and some other variables. Furthermore, because every dimension that is dependent on $a_t$ is necessary for the policy learning and every dimension that is (conditionally) independent of $a_t$ for at least a subset of other variables is not necessary for the policy learning, compact domain-specific $\boldsymbol{\theta}_k^{min}$ and compact shared representations $\mathbf{s}_t^{min}$ contain minimal and sufficient dimensions for policy learning across domains. Note that the agents determine the action under the condition of maximizing cumulative reward, which policy learning aims to achieve, so we always consider the situation when the discounted cumulative future reward $R_{t+1}$ is given. $\square$

## A2   PROOF OF THEOREM 1

**Theorem 1** (Structural Identifiability). *Suppose the underlying states $\mathbf{s}_t$ are observed, i.e., Eq. (1) is an MDP. Then under the Markov condition and faithfulness assumption, the structural matrices $\mathbf{C}^{\mathbf{s} \rightarrow \mathbf{s}}, \mathbf{c}^{a \rightarrow \mathbf{s}}, \mathbf{c}^{\mathbf{s} \rightarrow r}, c^{a \rightarrow r}, \mathbf{C}^{\theta_k \rightarrow \mathbf{s}},$ and $c^{\theta_k \rightarrow r}$ are identifiable .*

*Proof.* We concatenate data from different domains and denote by $k$ be the variable that takes the domain index $1, \cdots, n$. Since the data distribution changes across domains and the change is due to the unobserved change factors $\boldsymbol{\theta}_k$ that influence the observed variables, we can represent the change factors as a function of $k$. In other words, we use the domain index $k$ as a surrogate variable to characterize the unobserved change factors.

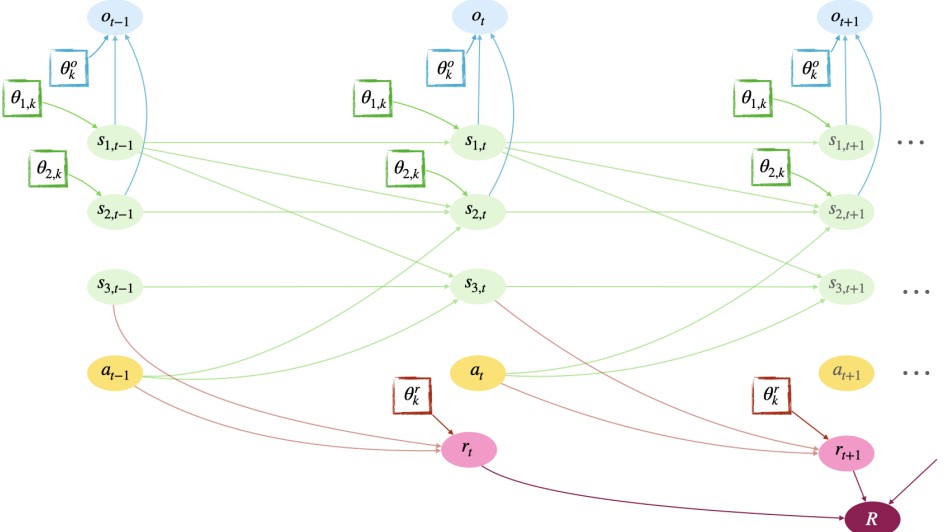

(a) An example of a ground Bayesian network (unrolled DBN over time).

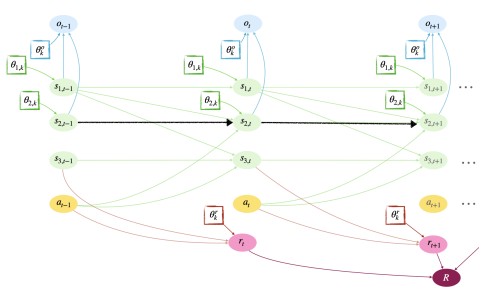

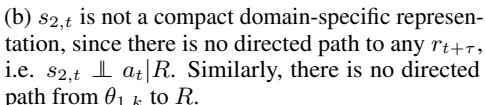

(b) $s_{2,t}$ is not a compact domain-specific representation, since there is no directed path to any $r_{t+\tau}$, i.e. $s_{2,t} \perp\!\!\!\perp a_t|R$. Similarly, there is no directed path from $\theta_{1,k}$ to $R$.

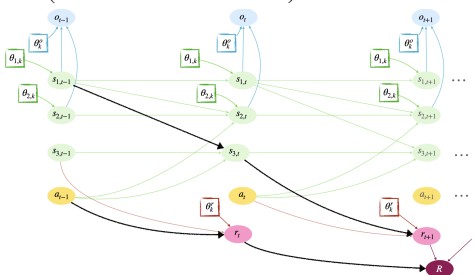

(c) $s_{1,t}$ is a compact domain-specific representation, since there exists a path to $a_t$ that is d-connected when we condition on the collider $R$. Similarly $\theta_{1,k}$ is d-connected to $a_t$ when conditioning on $R$.

Figure A2: Example model in which $s_{1,t}$ and $s_{3,t}$ are compact domain-specific representations for policy learning, while $s_{2,t}$ is not. This does not mean that $s_{2,t}$ and $\theta_{2,k}$ are not useful in the model estimation part, especially in estimating $\theta_k^o$.

We denote the variable set in the system by $\mathbf{V}$, with $\mathbf{V} = \{s_{1,t-1}, \ldots, s_{d,t-1}, s_{1,t}, \ldots, s_{d,t}, a_{t-1}, r_t\}$, and the variables form a dynamic Bayesian network $\mathcal{G}$. Note that in our particular problem, according to the generative environment model in Eq. (1), the possible edges in $\mathcal{G}$ are only those from $s_{i,t-1} \in \mathbf{s}_{t-1}$ to $s_{j,t} \in \mathbf{s}_t$, from $a_{t-1}$ to $s_{j,t} \in \mathbf{s}_t$, from $s_{i,t-1} \in \mathbf{s}_{t-1}$ to $r_t$, and from $a_{t-1}$ to $r_t$. We further include the domain index $k$ into the system to characterize the unobserved change factors.

It has been shown that under the Markov condition and faithfulness assumption, for every $V_i, V_j \in \mathbf{V}$, $V_i$ and $V_j$ are not adjacent in $\mathcal{G}$ if and only if they are independent conditional on some subset of $\{V_l | l \neq i, l \neq j\} \cup k$ (Huang et al., 2020). Thus, we can asymptotically identity the correct graph skeleton over $\mathbf{V}$.

Moreover, since we assume a dynamic Bayesian network, there the direction of an edge between a variable at time $t$ to one at time $t+1$ is fixed. Therefore, the structural matrices $\mathbf{C}^{\mathbf{s} \to \mathbf{s}}, \mathbf{c}^{a \to \mathbf{s}}, \mathbf{c}^{\mathbf{s} \to r}$, and $c^{a \to r}$, which are parts of the graph $\mathcal{G}$ over $\mathbf{V}$, are identifiable.

Furthermore, we want to show the identifiability of $\mathbf{C}^{\theta_k \to \mathbf{s}}$, and $c^{\theta_k \to r}$; that is, to identify which distribution modules have changes. Whether a variable $V_i$ has a changing module is decided by whether $V_i$ and $k$ are independent conditional on some subset of other variables. The justification for one side of this decision is trivial. If $V_i$'s module does not change, that means $P(V_i | \mathrm{PA}^i)$ remains the same for every value of $k$, and so $V_i \perp\!\!\!\perp k | \mathrm{PA}^i$. Thus, if $V_i$ and $k$ are not independent conditional on any subset of other variables, $V_i$'s module changes with $k$, which is represented by an edge between $V_i$ and $k$. Conversely, we assume that if $V_i$'s module changes, which entails that $V_i$ and $k$ are not independent given $\mathrm{PA}^i$, then $V_i$ and $k$ are not independent given any other subset of $\mathbf{V} \setminus \{V_i\}$. If this assumption does not hold, then we only claim to detect some (but not necessarily all) variables with changing modules. $\qquad \square$

## A3 PROOF OF THEOREM 2

In this section, we derive the generalization bound under the PAC-Bayes framework (McAllester, 1999; Shalev-Shwartz and Ben-David, 2014), and our formulation follows (Pentina and Lampert, 2014) and (Amit and Meir, 2018). We assume that all domains share the sample space $\mathcal{Z}$, hypothesis space $\mathcal{H}$, and loss function $\ell : \mathcal{H} \times \mathcal{Z} \to [0, 1]$. All domains differ in the unknown sample distribution $E_k$ parameterized by $\boldsymbol{\theta}_k^{\min}$ associated with each domain $k$. We observe the training sets $S_1, \ldots, S_n$ corresponding to $n$ different domains. The number of samples in domain $k$ is denoted by $m_k$. Each dataset $S_k$ is assumed to be generated from an unknown sample distribution $E_k^{m_k}$. We also assume that the sample distribution $E_k$ are generated $i.i.d.$ from an unknown domain distribution $\tau$. More specifically, we have $S_k = (z_{1,k}, \ldots, z_{i,k}, \ldots, z_{m_k,k})$, where $z_{i,k} = (\mathbf{s}_{i,k}, v^*(\mathbf{s}_{i,k}))$. Note that, $\mathbf{s}_{i,k}$ is the $i$-th state sampled from $k$-th domain and $v^*(\mathbf{s}_{i,k})$ is its corresponding optimal state-value. For any value function $h_{\boldsymbol{\theta}_k^{\min}}(\cdot)$ parameterized by $\boldsymbol{\theta}_k^{\min}$, we define the loss function $\ell(h_{\boldsymbol{\theta}_k^{\min}}, z_{i,k}) = D_{dist}(h_{\boldsymbol{\theta}_k^{\min}}(\mathbf{s}_{i,k}), v^*(\mathbf{s}_{i,k}))$, where $D_{dist}$ is a distance function that measures the discrepancy between the learned value and the optimal state-value. We also let $P$ be the prior distribution over $\mathcal{H}$ and $Q$ the posterior over $\mathcal{H}$.

**Theorem 1.** *Let $\mathcal{Q}$ be an arbitrary distribution over $\boldsymbol{\theta}_k^{min}$ and $\mathcal{P}$ the prior distribution over $\boldsymbol{\theta}_k^{min}$. Then for any $\delta \in (0, 1]$, with probability at least $1 - \delta$, the following inequality holds uniformly for all $\mathcal{Q}$,*

$$er(\mathcal{Q}) \leq \frac{1}{n} \sum_{k=1}^n \hat{er}(\mathcal{Q}, S_k) + \frac{1}{n} \sum_{k=1}^n \sqrt{\frac{1}{2(m_k - 1)} \left( D_{KL}(\mathcal{Q}||\mathcal{P}) + \log \frac{2nm_k}{\delta} \right)}$$
$$+ \sqrt{\frac{1}{2(n-1)} \left( D_{KL}(\mathcal{Q}||\mathcal{P}) + \log \frac{2n}{\delta} \right)},$$

*where $er(\mathcal{Q})$ and $\hat{er}(\mathcal{Q}, S_k)$ are the generalization error and the training error between the estimated value and the optimal value, respectively.*

*Proof.* This proof consists of two steps, both using the classical PAC-Bayes bound (McAllester, 1999; Shalev-Shwartz and Ben-David, 2014). Therefore, we start by restating the classical PCA-Bayes bound.

**Theorem A1** (Classical PAC-Bayes Bound, General Notations). *Let $\mathcal{X}$ be a sample space, $P(X)$ a distribution over $\mathcal{X}$, $\Theta$ a hypothesis space. Given a loss function $\ell(\theta, X) : \Theta \times \mathcal{X} \rightarrow [0, 1]$ and a collection of M i.i.d random variables $(X_1, \ldots, X_M)$ sampled from $P(X)$, let $\pi$ be a prior distribution over hypothesis in $\Theta$. Then, for any $\delta \in (0, 1]$, the following bound holds uniformly for all posterior distributions $\rho$ over $\Theta$,*

$$P\left(\mathop{\mathbb{E}}_{X_i \sim P(X), \theta \sim \rho}[\ell(\theta, X_i)] \le \frac{1}{M}\sum_{m=1}^{M}\mathop{\mathbb{E}}_{\theta \sim \rho}[\ell(\theta, X_m)] + \sqrt{\frac{1}{2(M-1)}\left(D_{KL}(\rho||\pi) + \log\frac{M}{\delta}\right)}, \forall \rho\right)$$
$$\ge 1 - \delta.$$

**Between-domain Generalization Bound**  First, we bound the between-domain generalization, i.e., relating $er(\mathcal{Q})$ to $er(\mathcal{Q}, E_k)$.

We first expand the generalization error as below,

$$\begin{aligned}
er(\mathcal{Q}) &= \mathop{\mathbb{E}}_{(E,m)\sim\tau} \mathop{\mathbb{E}}_{S\sim E^m} \mathop{\mathbb{E}}_{\boldsymbol{\theta}\sim\mathcal{Q}} \mathop{\mathbb{E}}_{h\sim Q(S,\boldsymbol{\theta})} \mathop{\mathbb{E}}_{z\sim E}\ell(h,z) \\
&= \mathop{\mathbb{E}}_{(E,m)\sim\tau} \mathop{\mathbb{E}}_{S\sim E^m} \mathop{\mathbb{E}}_{\boldsymbol{\theta}\sim\mathcal{Q}}\ell(\boldsymbol{\theta},E) \\
&= \mathop{\mathbb{E}}_{(E,m)\sim\tau} \mathop{\mathbb{E}}_{S\sim E^m} er(\mathcal{Q},E). 
\end{aligned} \tag{A1}$$

Then we compute the error across the training domains,

$$\frac{1}{n}\sum_{k=1}^{n}\mathop{\mathbb{E}}_{\boldsymbol{\theta}\sim\mathcal{Q}} \mathop{\mathbb{E}}_{h\sim Q(S_k,\boldsymbol{\theta})} \mathop{\mathbb{E}}_{z\sim E_k}\ell(h,z) = \frac{1}{n}\sum_{k=1}^{n}er(\mathcal{Q},E_k). \tag{A2}$$

Then Theorem A1 says that for any $\delta_0 \sim (0, 1]$, we have

$$P\left(er(\mathcal{Q}) \le \frac{1}{n}\sum_{k=1}^{n}er(\mathcal{Q},E_k) + \sqrt{\frac{1}{2(n-1)}\left(D_{KL}(\mathcal{Q}||\mathcal{P}) + \log\frac{n}{\delta_0}\right)}\right) \ge 1 - \delta_0, \tag{A3}$$

where $\mathcal{P}$ is a prior distribution over $\boldsymbol{\theta}$.

**Within-domain Generalization Bound**  Then, we bound the the within-domain generalization, i.e., relating $er(\mathcal{Q}, E_k)$ to $\hat{er}(\mathcal{Q}, S_k)$.

We first have

$$er(\mathcal{Q},E_k) = \mathop{\mathbb{E}}_{\boldsymbol{\theta}\sim\mathcal{Q}} \mathop{\mathbb{E}}_{h\sim Q(S_k,\boldsymbol{\theta})} \mathop{\mathbb{E}}_{z\sim E_k}\ell(h,z). \tag{A4}$$

Then we compute the empirical error across the training domains,

$$\hat{er}(\mathcal{Q},S_k) = \frac{1}{m_k}\sum_{j=1}^{m_k}\mathop{\mathbb{E}}_{h\sim Q(S_k,\boldsymbol{\theta})} \mathop{\mathbb{E}}_{z\sim E_k}\ell(h,z_{i,j}). \tag{A5}$$

According to Theorem A1, for any $\delta_D \sim (0, 1]$, we have

$$P\left(er(\mathcal{Q},E_k) \le \hat{er}(\mathcal{Q},S_k) + \sqrt{\frac{1}{2(m_k-1)}\left(D_{KL}(\rho||\pi) + \log\frac{m_k}{\delta_k}\right)}\right) \ge 1 - \delta_k. \tag{A6}$$

With the choice of $\pi = \int \mathcal{P}(\boldsymbol{\theta})Q(S_D, \boldsymbol{\theta})d\boldsymbol{\theta}$ and $\rho = \int \mathcal{Q}(\boldsymbol{\theta})Q(S_D, \boldsymbol{\theta})d\boldsymbol{\theta}$, we have that $D_{KL}(\rho||\pi) \le D_{KL}(\mathcal{Q}||\mathcal{P})$ (Yin et al., 2019). Thus, the above inequality can be further written as,

$$P\left(er(\mathcal{Q},E_k) \le \hat{er}(\mathcal{Q},S_k) + \sqrt{\frac{1}{2(m_k-1)}\left(D_{KL}(\mathcal{Q}||\mathcal{P}) + \log\frac{m_k}{\delta_k}\right)}\right) \ge 1 - \delta_k. \tag{A7}$$

**Overall Generalization Bound**    Combining Eq. (A3) and (A7) using the union bound and choosing that for any $\delta > 0$, set $\delta_0 \doteq \frac{\delta}{2}$ and $\delta_k \doteq \frac{\delta}{2n}$ for $k = 1, \ldots, n$, then we finally obtain,

$$
P\left( er(\mathcal{Q}) \le \frac{1}{n}\sum_{k=1}^{n} \hat{er}(\mathcal{Q}, S_k) + \frac{1}{n}\sum_{k=1}^{n} \sqrt{ \frac{1}{2(m_k - 1)}\left( D_{KL}(\mathcal{Q}||\mathcal{P}) + \log\frac{2nm_k}{\delta} \right) } \right.
$$
$$
\left. + \sqrt{ \frac{1}{2(n-1)}\left( D_{KL}(\mathcal{Q}||\mathcal{P}) + \log\frac{2n}{\delta} \right) } \right) \ge 1 - \delta. \quad \text{(A8)}
$$

$\square$

## A4    MORE DETAILS FOR MODEL ESTIMATION

### A4.1    LOCATING MODEL CHANGES

In real-world scenarios, it is often the case that changes to the environment are sparse and localized. Instead of assuming every function to change arbitrarily, which is inefficient and unnecessarily complex, we first identify possible locations of the changes. To this end, we concatenate data from different domains and denote by $k$ the variable that takes distinct values $1, \cdots, n$ to represent the domain index. Then, we exploit (conditional) independencies/dependencies to locate model changes. These (conditional) independencies/dependencies can be tested by kernel-based conditional independence tests (Zhang et al., 2011), which allows for both linear or nonlinear relationships between variables. Below we show that in some cases, we can identify the location of $\boldsymbol{\theta}_k$, by using the conditional independence relationships from concatenated data.

**Proposition A1.** *In POMDP, where the underlying states are latent, we can localize the changes by conditional independence relationships from concatenated observed data $\mathcal{D}$ in the following cases:*

$C_1$: *if $o_t \perp\!\!\!\perp k$, then there is neither a change in the observation function nor in the state dynamics for any state that is an input to the observation function;*

$C_2$: *if $o_t \perp\!\!\!\perp k$ and $a_t \not\perp\!\!\!\perp k|r_{t+1}$, then there is only a change in the reward function;*

$C_3$: *if $a_t \perp\!\!\!\perp k|r_{t+1}$, then there is neither a change in the reward function nor in the state dynamics for any state in $\mathbf{s}^{min}$;*

$C_4$: *if $a_t \perp\!\!\!\perp k|r_{t+1}$ and $o_t \not\perp\!\!\!\perp k$, then there is a change in the observation function, or there exists a state that is not in $\mathbf{s}^{min}$ but is an input to the observation function, whose dynamics has a change.*

*Proof.* We first formulate the problem as follows. We concatenate data from different domains and use domain-index variable $k$ to indicate whether the corresponding distribution module has changes across domains. Specifically, by assuming the Markov condition and faithfulness assumption, $s_{i,t}$ has an edge with $k$ if and only if $p(s_{i,t}|PA(s_{i,t}))$ changes across domains, where $PA(\cdot)$ denotes its parents. Similarly, $r_t$ has an edge with $k$ if and only if $p(r_t|PA(r_t))$ changes across domains, and $o_t$ has an edge with $k$ if and only if $p(o_t|PA(o_t))$ changes across domains. Under this setting, locating changes is equivalent to identify which variables have an edge with $k$ from the data.

We consider the scenario of POMDP, where we only observe $\{o_t, r_t, a_t\}$ and the underlying states $\mathbf{s}_t$ are not observed. Below, we consider each case separately.

Case 1: Show that if $o_t \perp\!\!\!\perp k$, then there is neither a change in the observation function nor a change in the state dynamics for any state that is an input to the observation function.

We prove it by contradiction. Suppose that there is a change in the observation function and a change in the state dynamics for any state that is an input to the observation function. That is, $o_t$ has an edge with $k$, and $s_{i,t}$ that has a direct edge to $o_t$ also connects with $k$. Based on faithfulness assumption, $o_t \not\perp\!\!\!\perp k$, which contradicts to the assumption. Since we have a contradiction, it must be that there is neither a change in the observation function nor a change in the state dynamics for any state that is an input to the observation function.

Case 2: Show that if $o_t \perp\!\!\!\perp k$ and $a_t \not\perp\!\!\!\perp k|r_{t+1}$, then there is only a change in the reward function.

If $o_t \perp\!\!\!\perp k$ and $a_{t-1} \not\perp\!\!\!\perp k | r_t$, based on the Markov condition and faithfulness assumption, $r_t$ has an edge with $k$, and $s_{i,t}$ and $o_t$ do not have edges with $k$; that is, there are only changes in the reward function.

Case 3: Show that if $a_t \perp\!\!\!\perp k | r_{t+1}$, then there is neither a change in the reward function nor a change in the state dynamics for any state in $\mathbf{s}^{min}$.

By contradiction, suppose that there is a change in the reward function or there exists a state $s_{j,t} \in \mathbf{s}_j^{min}$ that has a change in its dynamics. That is, $r_t$ has an edge with $k$ or corresponding $s_{j,t}$ has an edge with $k$. Based on faithfulness assumption, $a_t \not\perp\!\!\!\perp k | r_{t+1}$, which contradicts to the assumption.

Case 4: Show that if $a_t \perp\!\!\!\perp k | r_{t+1}$ and $o_t \not\perp\!\!\!\perp k$, then there is a change in the observation function, or there exists a state that is not in $\mathbf{s}^{min}$ but is an input to the observation function, whose dynamics has a change.

According to Case 3, if $a_t \perp\!\!\!\perp k | r_{t+1}$, then there is neither a change in the reward function nor a change in the state dynamics for any state in $\mathbf{s}^{min}$. Furthermore, since $o_t \not\perp\!\!\!\perp k$, then based on the Markov condition, either $o_t$ has an edge with $k$, or there exists a state that is not in $\mathbf{s}^{min}$ but is an input to the observation function, whose dynamics has an edge with $k$. That is, there is a change in the observation function, or there exists a state that is not in $\mathbf{s}^{min}$ but is an input to the observation function, whose dynamics has a change.

$\square$

Based on Theorem 1 and Proposition A1, in MDP, we can fully determine where the changes are, so we only need to consider the corresponding $\theta_k^{(\cdot)}$ to capture the changes. In POMDP, in Case 1, we only need to involve $\theta_k^s$ and $\theta_k^r$ in model estimation, that is, $\boldsymbol{\theta}_k = \{\theta_k^s, \theta_k^r\}$; in Case 2, $\boldsymbol{\theta}_k = \{\theta_k^r\}$; and in Case 3 & 4, $\boldsymbol{\theta}_k = \{\theta_k^o, \theta_k^s\}$. For other cases, we involve $\boldsymbol{\theta}_k = \{\theta_k^o, \theta_k^s, \theta_k^r\}$ in model estimation.

### A4.2 MORE DETAILS FOR ESTIMATION OF DOMAIN-VARYING MODELS IN ONE STEP

We use MiSS-VAE to learn the environment model, which contains three components: the "sequential VAE" component, the "multi-model" component, and the "structure" component. Figure A3 gives the diagram of neural network architecture in model training.

Specifically, for the "sequential VAE" component, we include a Long Short-Term Memory (LSTM (Hochreiter and Schmidhuber, 1997)) to encode the sequential information with output $h_t$ and a Mixture Density Network (MDN (Bishop, 1994)) to output the parameters of MoGs, and thus to learn the inference model $q_\phi(\mathbf{s}_{t,k} | \mathbf{s}_{t-1,k}, \mathbf{y}_{1:t,k}, a_{1:t-1,k}; \boldsymbol{\theta}_k)$ and infer a sample of $\mathbf{s}_{t,k}$ from $q_\phi$ as the output. The generated sample further acts as an input to the decoder, and the decoder outputs $\hat{o}_{t+1}$ and $\hat{r}_{t+2}$. Moreover, the state dynamics which satisfies a Markov process is modeled with an MLP and MDN.

For the "multi-model" component, we include the domain index $k$ as an input to LSTM and involve $\boldsymbol{\theta}_k$ as free parameters in the inference model $q_\phi$, by assuming that $\boldsymbol{\theta}_k$ also characterizes the changes in the inference model. Moreover, we embed $\theta_k^s$ in state dynamics $p_\gamma$, $\theta_k^o$ in observation function and $\theta_k^r$ in reward function in the decoder. With such a characterization, except $\boldsymbol{\theta}_k$, all other parameters are shared across domains, so that all we need to update in the target domain is the low-dimensional $\boldsymbol{\theta}_k$, which greatly improves the sample efficiency and the statistical efficiency in the target domain–usually few samples are needed.

For the "structure" component, the latent states are organized with structures, captured by the mask $\mathbf{C}^{\mathbf{s} \to \mathbf{s}}$. Also, the structural relationships among perceived signals, latent states, the action variable, the reward variable, and the domain-specific factors are embedded as free parameters (structural vectors and scalars $\mathbf{c}^{\mathbf{s} \to \mathbf{s}}$, $c^{a \to \mathbf{s}}$, $\mathbf{c}^{\theta_k \to \mathbf{s}}$, $\mathbf{c}^{\mathbf{s} \to r}$, and $c^{a \to r}$) into MiSS-VAE.

## A5 COMPLETE EXPERIMENTAL RESULTS

In this section we provide the complete experimental results on both of our settings, modified Cartpole and modified Pong in the OpenAI Gym (Brockman et al., 2016). In the POMDP setting, we use images as input, which for Cartpole look like Fig. A4(a) and for Pong look like Fig. A8(a). For

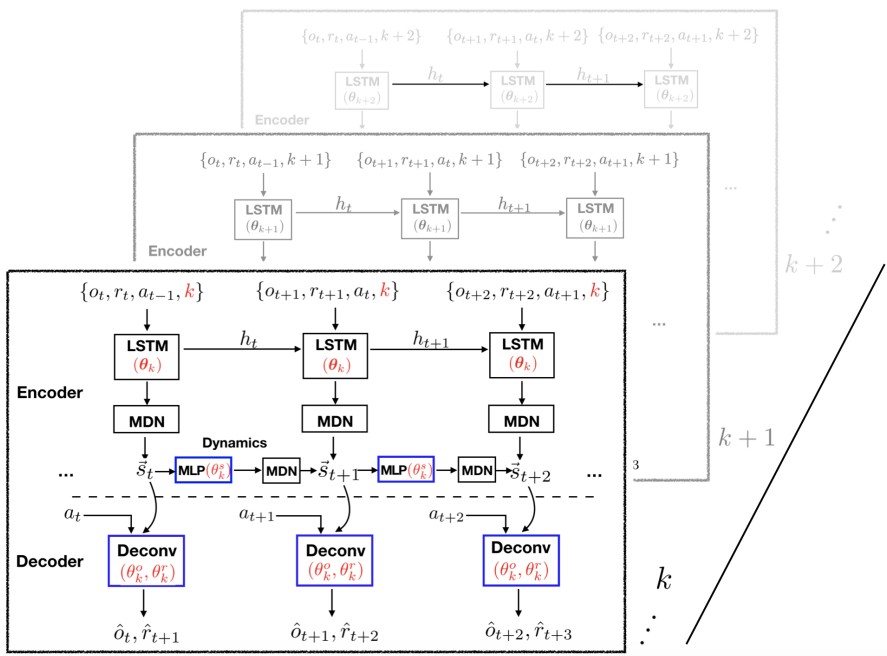

Figure A3: Diagram of MiSS-VAE neural network architecture. The "sequential VAE" component, "multi-model" component, and "structure" component are marked with black, red, and blue, respectively.

Cartpole, we also consider the MDP case, where the true states (cart position and velocity, pole angle and angular velocity) are used as the input to the model.

We consider changes in the state dynamics (e.g., the change of gravity or cart mass in Cartpole, or the change of orientation in Atari), changes in perceived signals (e.g., different noise levels on observed images in Cartpole, as shown in Fig. A4 or colors in Pong) and changes in reward functions (e.g., different reward functions in Pong based on the contact point of the ball), as shown in Fig. A8 for Pong. For each of these factors, we take into account both *interpolation* (where the factor value in the target domain is in the support of that in source domains), and *extrapolation* (where it is out of the support w.r.t. the source domains).

We train on $n$ source domains based on the trajectory data generated by a random policy. In Cartpole experiments, for each domain we collect 10000 trials with 40 steps. For Pong experiments, each domain contains 40 episodes data and each of them takes a maximum of 10000 steps.

## A5.1 COMPLETE RESULTS OF MODIFIED CARTPOLE EXPERIMENT

The Cartpole problem consists of a cart and a vertical pendulum attached to the cart using a passive pivot joint. The cart can move left or right. The task is to prevent the vertical pendulum from falling by putting a force on the cart to move it left or right. The action space consists of two actions: moving left or right.

We introduce two change factors for the state dynamics $\theta_k^s$: varying gravity and varying mass of the cart, and a change factor in the observation function $\theta_k^o$ that is the image noise level. Fig. A4 gives a visual example of Cartpole game, and the image with Gaussian noise. The images of the varying gravity and mass look exactly like the original image. Specifically, in the gravity case, we consider source domains with gravity $g = \{5, 10, 20, 30, 40\}$. We take into account both interpolation (where the gravity in the target domain is in the support of that in source domains) with $g = \{15\}$, and extrapolation (where it is out of the support w.r.t. the source domains) with $g = \{55\}$. Similarly, we consider source domains where the mass of the cart is $m = \{0.5, 1.5, 2.5, 3.5, 4.5\}$, while in target domains it is $m = \{1.0, 5.5\}$. In terms of changes on the observation function $\theta_k^o$, we add

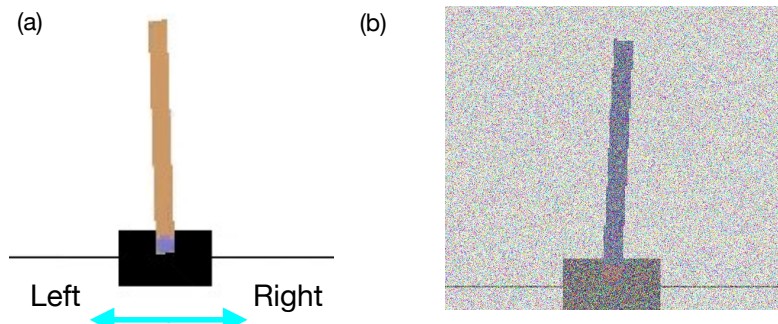

Figure A4: Visual examples of Cartpole game and change factors. (a) Cartpole game; (b) Modified Cartpole game with Gaussian noise on the image. The light blue arrows are added to show the direction in which the agent can move.

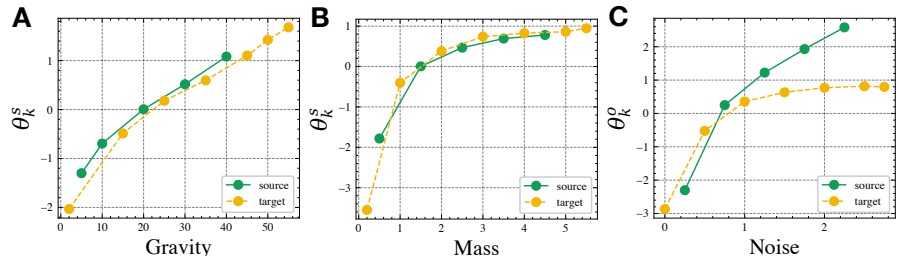

Figure A5: The estimated $\boldsymbol{\theta}_k^{min}$ for the three change factors in Cartpole (POMDP): gravity (A), mass (B), and noise level (C) for $N_{target} = 50$. .

Gaussian noise on the images with variance $\sigma = \{0.25, 0.75, 1.25, 1.75, 2.25\}$ in source domains, and $\sigma = \{0.5, 2.75\}$ in target domains.

We summarize the detailed settings in both source and target domains in Table A1. In particular in each experiment we use all source domains for each change factor and one of the target domains at a time in either the interpolation and extrapolation set.

| | Gravity | Mass | Noise |
|---|---|---|---|
| Source domains | $\{5, 10, 20, 30, 40\}$ | $\{0.5, 1.5, 2.5, 3.5, 4.5\}$ | $\{0.25, 0.75, 1.25, 1.75, 2.25\}$ |
| Interpolation set | $\{15\}$ | $\{1.0\}$ | $\{0.5\}$ |
| Extrapolation set | $\{55\}$ | $\{5.5\}$ | $\{2.75\}$ |

Table A1: The settings of source and target domains for modified Cartpole experiments.

### A5.1.1 LEARNED $\theta_k$ IN CARTPOLE EXPERIMENTS

Fig. A5 shows the estimated $\boldsymbol{\theta_k}$ in the modified Cartpole experiments. For the gravity and mass scenarios, the learned parameters with different sample sizes are close with each other. This phenomenon indicates that even with only a few samples ($N_{target} = 50$), AdaRL can estimate these change parameters very well. For the noise level factor, the learned curves with different sample sizes have a similar behaviour, but the distance is larger. We can see that the $\theta_k$ we learn is approximately a linear function of the actual perturbation in gravity, while for the mass and noise are monotonic functions.

### A5.1.2 AVERAGE FINAL SCORES FOR MULTIPLE $N_{target}$ IN CARTPOLE EXPERIMENTS

Tables A2 and A3 show the complete results of the modified Cartpole experiments (POMDP settings) for $N_{target} = 20$ and $N_{target} = 10000$. Table A4 and A5 give the complete results of the modified

| | Oracle
Upper bound | Non-t
lower bound | PNN
(Rusu et al., 2016) | PSM
(Agarwal et al., 2021a) | MTQ
(Fakoor et al., 2020) | AdaRL*
Ours w/o masks | AdaRL
Ours |
|---|---|---|---|---|---|---|---|
| G_in | 1930.5
($\pm$1042.6) | 828.5 $\bullet$
($\pm$509.4) | 1113.4 $\bullet$
($\pm$719.2) | 1008.5 $\bullet$
($\pm$453.6) | 1257.2
($\pm$503.5) | 1290.6
($\pm$589.1) | 1302.7
($\pm$874.0) |
| G_out | 408.6
($\pm$67.2) | 54.0 $\bullet$
($\pm$13.6) | 109.7 $\bullet$
($\pm$24.2) | 156.8 $\bullet$
($\pm$49.5) | 120.5 $\bullet$
($\pm$87.4) | 173.8
($\pm$39.3) | 198.4
($\pm$54.5) |
| M_in | 2004.8
($\pm$404.3) | 447.2 $\bullet$
($\pm$39.6) | 1120.6 $\bullet$
($\pm$348.1) | 982.5 $\bullet$
($\pm$363.2) | 1245.7 $\bullet$
($\pm$274.0) | 1095.8 $\bullet$
($\pm$521.3) | **1361.0**
($\pm$327.3) |
| M_out | 1498.6
($\pm$625.4) | 130.6 $\bullet$
($\pm$39.8) | 528.5 $\bullet$
($\pm$251.4) | 830.6 $\bullet$
($\pm$317.2) | 875.2 $\bullet$
($\pm$262.5) | 764.2 $\bullet$
($\pm$320.9) | **1082.5**
($\pm$236.3) |
| N_in | 8640.5
($\pm$3086.1) | 679.4 $\bullet$
($\pm$283.5) | 4170.6 $\bullet$
($\pm$2202.2) | 4936.5 $\bullet$
($\pm$1604.9) | 3985.7 $\bullet$
($\pm$2387.4) | 4954.3 $\bullet$
($\pm$2627.8) | **5761.2**
($\pm$2341.5) |
| N_out | 4465.2
($\pm$667.3) | 584.0 $\bullet$
($\pm$429.2) | 2841.5 $\bullet$
($\pm$385.2) | 2650.2 $\bullet$
($\pm$453.6) | 2654.0 $\bullet$
($\pm$277.9) | 1785.2 $\bullet$
($\pm$470.3) | **3318.7**
($\pm$293.5) |

Table A2: Average final scores in modified Cartpole (POMDP) with $N_{target} = 20$. The best non-oracle results are marked in red. G, M, and N denote the gravity, mass, and noise respectively.

| | Oracle
Upper bound | Non-t
lower bound | PNN
(Rusu et al., 2016) | PSM
(Agarwal et al., 2021a) | MTQ
(Fakoor et al., 2020) | AdaRL*
Ours w/o masks | AdaRL
Ours |
|---|---|---|---|---|---|---|---|
| G_in | 1930.5
($\pm$1042.6) | 1115.2 $\bullet$
($\pm$341.8) | 1637.4 $\bullet$
($\pm$378.2) | 1838.4
($\pm$358.1) | 1459.2 $\bullet$
($\pm$688.5) | 1864.5
($\pm$694.1) | 1924.6
($\pm$874.0) |
| G_out | 408.6
($\pm$67.2) | 161.3 $\bullet$
($\pm$65.9) | 329.6 $\bullet$
($\pm$48.9) | 457.3
($\pm$138.5) | 393.2 $\bullet$
($\pm$76.5) | 384.2 $\bullet$
($\pm$103.7) | 410.6
($\pm$92.3) |
| M_in | 2004.8
($\pm$404.3) | 596.0 $\bullet$
($\pm$373.4) | 1672.3 $\bullet$
($\pm$642.9) | 1798.5 $\bullet$
($\pm$493.0) | 1905.4
($\pm$378.2) | 1864.2
($\pm$309.5) | 1898.5
($\pm$683.4) |
| M_out | 1498.6
($\pm$625.4) | 325.6 $\bullet$
($\pm$146.3) | 1206.8 $\bullet$
($\pm$394.7) | 1339.4 $\bullet$
($\pm$520.5) | 1296.2 $\bullet$
($\pm$773.1) | 1297.4 $\bullet$
($\pm$411.2) | **1486.3**
($\pm$598.2) |
| N_in | 8640.5
($\pm$3086.1) | 1239.6 $\bullet$
($\pm$380.5) | 6476.2 $\bullet$
($\pm$3132.9) | 7493.4 $\bullet$
($\pm$1981.5) | 7932.9
($\pm$2389.0) | 7382.4 $\bullet$
($\pm$2915.3) | 8179.8
($\pm$2356.0) |
| N_out | 4465.2
($\pm$667.3) | 962.5 $\bullet$
($\pm$341.8) | 3043.9 $\bullet$
($\pm$1098.6) | 2987.2 $\bullet$
($\pm$1172.3) | 3892.4 $\bullet$
($\pm$763.0) | 4183.6
($\pm$782.2) | 4235.2
($\pm$532.4) |

Table A3: Average final scores in modified Cartpole (POMDP) with $N_{target} = 10000$. The best non-oracle results are marked in red. G, M, and N denote the gravity, mass, and noise respectively.

| | Oracle
Upper bound | Non-t
lower bound | CAVIA
(Zintgraf et al., 2019) | PEARL
(Rakelly et al., 2019) | AdaRL*
Ours w/o masks | AdaRL
Ours |
|---|---|---|---|---|---|---|
| G_in | 2486.1
($\pm$369.7) | 972.6 $\bullet$
($\pm$368.5) | 1651.5 $\bullet$
($\pm$623.8) | 1720.3 $\bullet$
($\pm$589.4) | 1602.7 $\bullet$
($\pm$393.6) | **1943.2**
($\pm$765.4) |
| G_out | 693.9
($\pm$100.6) | 243.8 $\bullet$
($\pm$45.2) | 356.2
($\pm$76.5) | 362.1
($\pm$57.3) | 292.4 $\bullet$
($\pm$91.8) | 395.6
($\pm$101.7) |
| M_in | 2678.2
($\pm$630.5) | 480.3 $\bullet$
($\pm$136.2) | 1306.8 $\bullet$
($\pm$376.5) | 1589.4 $\bullet$
($\pm$682.3) | 1624.8 $\bullet$
($\pm$531.6) | **1962.0**
($\pm$652.8) |
| M_out | 1405.6
($\pm$368.0) | 306.5 $\bullet$
($\pm$162.4) | 853.2 $\bullet$
($\pm$317.6) | 969.4 $\bullet$
($\pm$238.5) | 984.6 $\bullet$
($\pm$209.8) | **1113.5**
($\pm$394.2) |
| G_in
& M_in | 1984.2
($\pm$871.3) | 374.9 $\bullet$
($\pm$126.8) | 1174.3 $\bullet$
($\pm$298.2) | 964.3 $\bullet$
($\pm$370.5) | 1209.6 $\bullet$
($\pm$425.7) | **1392.7**
($\pm$392.6) |
| G_out
& M_out | 939.4
($\pm$270.5) | 292.4 $\bullet$
($\pm$127.6) | 494.6 $\bullet$
($\pm$201.3) | 368.4 $\bullet$
($\pm$259.8) | 399.8 $\bullet$
($\pm$242.5) | **531.2**
($\pm$272.5) |

Table A4: Average final scores in modified Cartpole (MDP) with $N_{target} = 20$. The best non-oracle results are marked in red, while **bold** indicates a statistically significant result w.r.t. all the baselines. G, M, and N denote the gravity, mass, and noise respectively. "$\bullet$" indicates the baselines for which the improvements of AdaRL are statistically significant (via Wilcoxon signed-rank test at $5\%$ significance level).

|  | Oracle
Upper bound | Non-t
lower bound | CAVIA
(Zintgraf et al., 2019) | PEARL
(Rakelly et al., 2019) | AdaRL*
Ours w/o masks | AdaRL
Ours |
|---|---|---|---|---|---|---|
| G_in | 2486.1
(±369.7) | 986.3 ●
(±392.5) | 1907.4 ●
(±526.8) | 2102.3 ●
(±398.5) | 1864.0 ●
(±369.2) | **2365.1**
(±403.5) |
| G_out | 693.9
(±100.6) | 349.2 ●
(±72.0) | 502.9 ●
(±133.2) | 585.7 ●
(±98.6) | 494.7 ●
(±151.4) | **604.8**
(±117.6) |
| M_in | 2678.2
(±630.5) | 643.9 ●
(±281.3) | 2008.6 ●
(±436.2) | 2106.2 ●
(±436.7) | 2148.9 ●
(±387.2) | **2415.2**
(±591.4) |
| M_out | 1405.6
(±368.0) | 617.4 ●
(±145.3) | 1182.7 ●
(±255.8) | 1294.5
(±210.6) | 1207.5
(±251.3) | 1263.5
(±362.9) |
| G_in
& M_in | 1984.2
(±871.3) | 452.6 ●
(±178.3) | 1275.0 ●
(±432.5) | 1468.7 ●
(±697.2) | 1395.4 ●
(±387.2) | **1589.4**
(±379.5) |
| G_out
& M_out | 939.4
(±270.5) | 596.2 ●
(±137.5) | 709.5 ●
(±386.0) | 743.8 ●
(±200.9) | 724.7
(±283.8) | 769.3
(±208.4) |

Table A5: Average final scores in modified Cartpole (MDP) with $N_{target} = 10000$. The best non-oracle results are marked in red, while **bold** indicates a statistically significant result w.r.t. all the baselines. G, M, and N denote the gravity, mass, and noise respectively.

Cartpole experiments (MDP settings with symbolic input) for $N_{target} = 20$ and $N_{target} = 10000$. We average the scores across 30 trials from different random seeds during the policy learning stage. The results suggest that, in most cases, AdaRL can outperform other baselines.

### A5.1.3 AVERAGE POLICY LEARNING CURVES IN TERMS OF STEPS

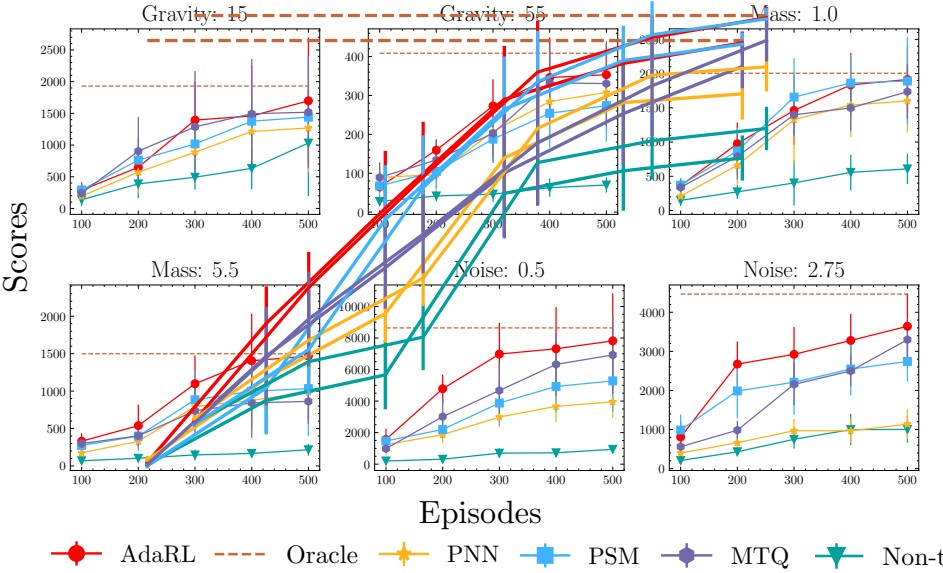

Figure A6: Learning curves for modified Cartpole experiments (POMDP version) with change factors. The reported scores are averaged across 30 trials.

Fig. A7 and A6 provide the learning curves for modified Cartpole experiments (MDP and POMDP versions) with multiple change factors. In most cases, AdaRL can converge faster than other baselines.

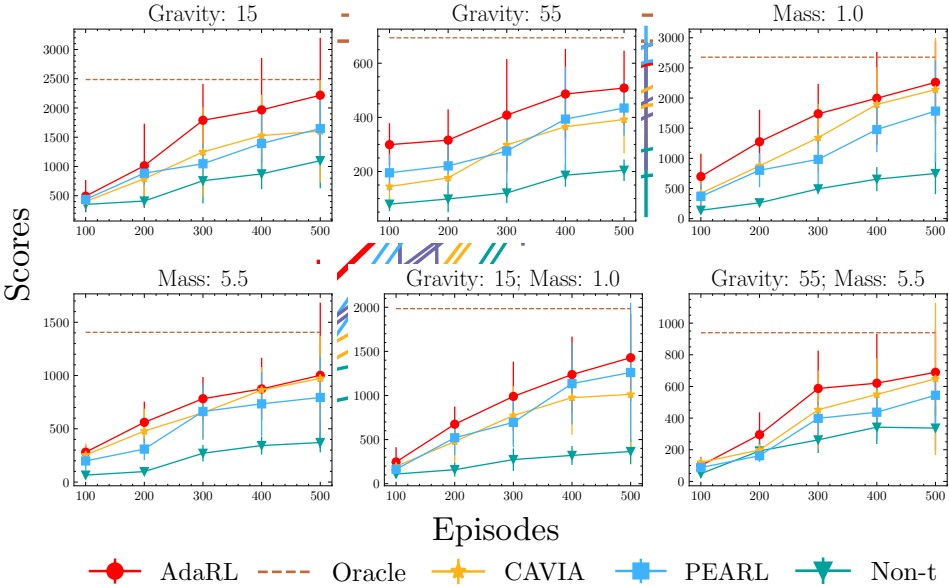

Figure A7: Learning curves for modified Cartpole experiments (MDP version) with change factors. The reported scores are averaged across 30 trials.

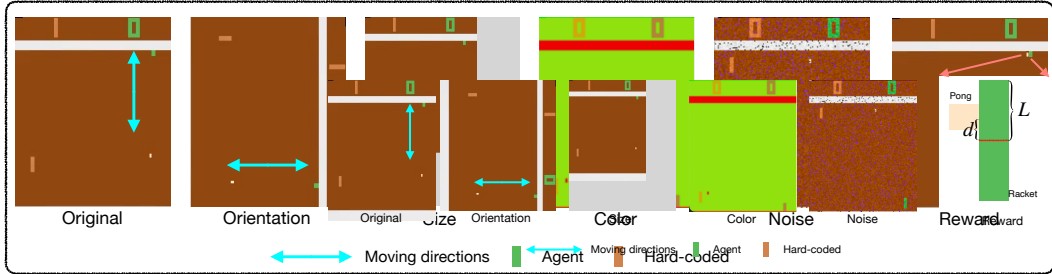

Figure A8: Visual example of the original Pong game and the various change factors. The light blue arrows are added to show the direction in which the agent can move.

### A5.2 COMPLETE RESULTS OF THE MODIFIED PONG EXPERIMENT WITH CHANGING DYNAMICS AND OBSERVATIONS

Atari Pong is a two-dimensional game that simulates table tennis. The agent controls a paddle moving up and down vertically, aiming at hitting the ball. The goal for the agent is to reach higher scores, which are earned when the other agent (hard-coded) fails to hit back the ball. We show the example of the original visual inputs and how it appears after we have changed each of the change factors in Fig. A8.

In source domains, the degrees are chosen from $\omega = \{0°, 180°\}$, and in target domains, they are chosen from $\omega = \{90°, 270°\}$. For the image size, we reduce the original image by a factor of $\{2, 4, 6, 8\}$ in source domains and by a factor of $\{3, 9\}$ in target domains.

We summarize the detailed settings in both source and target domains in Table A6. In particular, in each experiment we use all source domains for each change factor and one of the target domains at a time in either the interpolation and extrapolation set.

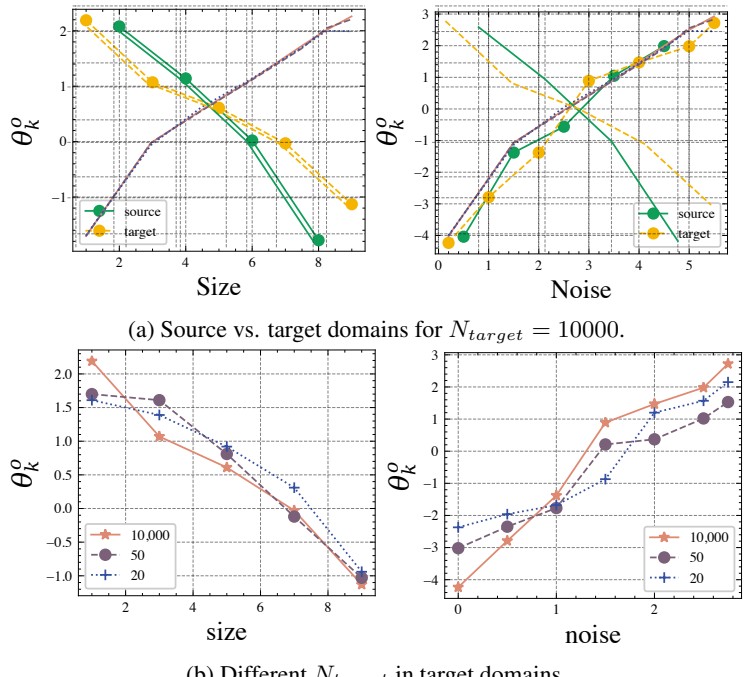

(a) Source vs. target domains for $N_{target} = 10000$.

(b) Different $N_{target}$ in target domains.

Figure A9: Learned $\theta_k^o$ for the two change factors, size and noise, in modified Pong.

|  | Size | Orientations | Noise | Background colors |
|---|---|---|---|---|
| Source domains | $\{2, 4, 6, 8\}$ | $0°, 180°$ | $\{0.25, 0.75, 1.25, 1.75, 2.25\}$ | original, green, red |
| Interpolation set | $\{3\}$ | $90°$ | $\{1.0\}$ | yellow |
| Extrapolation set | $\{9\}$ | $270°$ | $\{2.75\}$ | white |

Table A6: The settings of source and target domains for modified Pong experiments.

### A5.2.1 LEARNED $\theta_k$ IN MODIFIED PONG EXPERIMENTS

Fig. A9a, Table A7, and Table A8 show the learned $\theta_k$ in modified Pong experiments across different change factors.

Table A7: The learned $\theta_k^s$ across different orientation angles with different $N_{target}$ in modified Pong. The bold columns represent the target domains.

| $N_{target}$ | Orientations | $0°$ | **$90°$** | $180°$ | **$270°$** |
|---|---|---|---|---|---|
| 10000 | $\theta_k^{s1}$ | $-2.32$ | $-1.78$ | $1.69$ | $0.44$ |
|  | $\theta_k^{s2}$ | $-2.94$ | $-1.86$ | $1.47$ | $1.59$ |
| 50 | $\theta_k^{s1}$ | $-2.01$ | $-0.87$ | $1.85$ | $0.69$ |
|  | $\theta_k^{s2}$ | $-2.59$ | $-1.84$ | $1.42$ | $1.07$ |
| 20 | $\theta_k^{s1}$ | $-1.69$ | $-1.23$ | $0.79$ | $1.38$ |
|  | $\theta_k^{s2}$ | $-1.98$ | $-0.56$ | $0.82$ | $1.20$ |

We can find that each dimension of the learned $\boldsymbol{\theta}_k^s$ is a nonlinear monotonic function of the change factors. Table A7, Table A8 and Fig. A9b also give the learned $\theta^k$ with different sample sizes $N_{target}$ in target domains. Similarly, the learned curves with different sample sizes are homologous. Even with a few samples, AdaRL can still capture the model changes well.

Table A8: The learned $\theta_k^o$ across different colors with different $N_{target}$ in modified Pong. The bold columns represent the target domains.

| $N_{target}$ | Colors | Original | Red | Green | **Yellow** | **White** |
|---|---|---|---|---|---|---|
| 10000 | $\theta_k^{o1}$ | 1.36 | 1.47 | 1.04 | 1.58 | $-0.91$ |
| | $\theta_k^{o2}$ | 0.72 | $-1.15$ | 1.17 | 0.96 | $-1.33$ |
| | $\theta_k^{o3}$ | 0.93 | $-1.28$ | $-1.31$ | $-0.65$ | $-1.09$ |
| 50 | $\theta_k^{o1}$ | 0.96 | 1.13 | 0.82 | 1.26 | $-0.73$ |
| | $\theta_k^{o2}$ | 0.59 | $-0.46$ | 0.75 | 1.32 | $-0.59$ |
| | $\theta_k^{o3}$ | 0.61 | $-1.02$ | $-0.91$ | $-0.18$ | $-0.49$ |
| 20 | $\theta_k^{o1}$ | 1.09 | 1.38 | 0.65 | 1.30 | $-0.46$ |
| | $\theta_k^{o2}$ | 0.58 | $-0.72$ | 0.39 | 1.60 | $-0.27$ |
| | $\theta_k^{o3}$ | 0.38 | $-0.59$ | $-0.63$ | $-0.24$ | $-0.33$ |

### A5.2.2 AVERAGE FINAL SCORES FOR MULTIPLE $N_{target}$

Table A2 and A3 provides the complete results of the modified Pong experiments with $N_{target} = 20$ and 10000, respectively. The details of both source and target domains are listed in Table A6. Similar to the results of Cartpole, AdaRL can perform the best among all baselines in Pong experiments. As shown in the results of the main paper, AdaRL consistently outperforms the other methods.

### A5.2.3 AVERAGE POLICY LEARNING CURVES IN TERMS OF STEPS

Fig. A12 gives the learning curves for modified Pong experiments with multiple change factors. From the results, we can find that AdaRL can converge faster than other baselines.

### A5.3 COMPLETE RESULTS OF THE MODIFIED PONG EXPERIMENT WITH CHANGING REWARD FUNCTIONS

Table A9 summarizes the detailed change factors in both linear and non-linear reward groups.

| | Linear reward ($k_1$) | Non-linear reward ($k_2$) |
|---|---|---|
| Source domains | $\{0.1, 0.2, 0.3, 0.4, 0.6, 0.7, 0.8\}$ | $\{2.0, 3.0, 5.0, 6.0, 7.0, 8.0, 9.0\}$ |
| Interpolation set | $\{0.5\}$ | $\{4.0\}$ |
| Extrapolation set | $\{0.9\}$ | $\{1.0\}$ |

Table A9: The settings of source and target domains for modified Pong experiments.

We denote with $L$ the half-length of the paddle and then formulate the two groups of reward functions as: (1) Linear reward functions: $r_t = \frac{k_1 d}{L}$, where $k_1 \in \{0.1, 0.2, 0.3, 0.4, 0.6, 0.7, 0.8\}$ in source domains and $k_1 \in \{0.5, 0.9\}$ in target domains; and (2) Non-linear reward functions: $r_t = \frac{k_2 L}{d + 3L}$, where $k_2 \in \{2.0, 3.0, 5.0, 6.0, 7.0, 8.0, 9.0\}$ in source domains and $k_2 \in \{1.0, 4.0\}$ in target domains.

### A5.3.1 LEARNED $\theta_r$ IN MODIFIED PONG EXPERIMENTS

Fig. A10 and A11 give the learned $\theta_r$ with both linear and non-linear rewards. In both groups, the learned $\theta_r$ is linearly or monotonically correlated with the change factor $k_1$ and there is no significant gap between the learned $\theta_r$ with different $N_{target}$.

### A5.3.2 AVERAGE FINAL SCORES FOR MULTIPLE $N_{target}$

Table A12, A13 and A14 shows the average final scores for $N_{target} = 50$, 10000 and 50000 in modified Pong experiments with changing rewards. AdaRL consistently outperforms the other methods across different $N_{target}$.

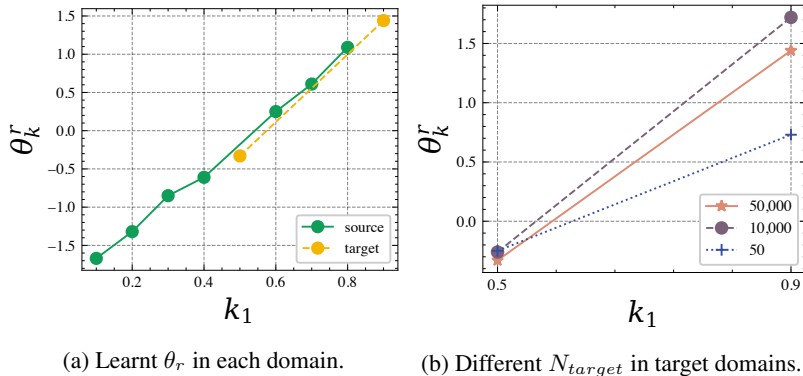

(a) Learnt $\theta_r$ in each domain.  (b) Different $N_{target}$ in target domains.

Figure A10: Learned $\theta_k^r$ for the linear changing rewards in modified Pong.

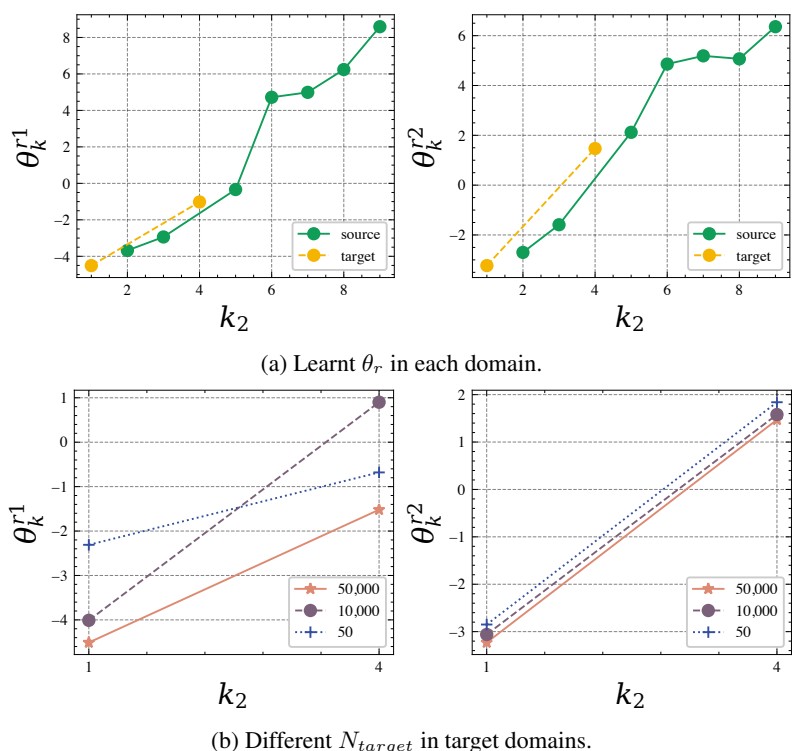

(a) Learnt $\theta_r$ in each domain.

(b) Different $N_{target}$ in target domains.

Figure A11: Learned $\theta_k^r$ for the non-linear changing rewards in modified Pong.

### A5.3.3 AVERAGE POLICY LEARNING CURVES IN TERMS OF STEPS

Fig. A12 (last two rows) gives the learning curves for modified Pong experiments with changing rewards.

### A5.4 RESULTS ON MUJOCO BENCHMARKS

We also apply AdaRL on MuJoCo benchmarks (Todorov et al., 2012), including Cheetah and Ant with a set of target velocities. We follow the setup in MAML (Finn et al., 2017) and CAVIA (Zintgraf et al., 2019). In model estimation stage, we choose 20 tasks for each of the game. The goal velocity of each task is sampled between 0.0 and 2.0 for the cheetah and between 0.0 and 3.0 for ant. The reward is the negative absolute value between agent's current and the goal velocity. In policy optimization stage, we utilize Trust Region Policy Optimization (TRPO Schulman et al. (2015)). The results on

|  | Oracle Upper bound | Non-t lower bound | PNN (Rusu et al., 2016) | PSM (Agarwal et al., 2021a) | MTQ (Fakoor et al., 2020) | AdaRL* Ours w/o masks | AdaRL Ours |
|---|---|---|---|---|---|---|---|
| O_in | 18.65 (±2.43) | 4.30 ● (±2.95) | 8.68 ● (±5.78) | 9.65 ● (±3.19) | 14.80 ● (±2.02) | 15.08 ● (±3.19) | **16.79** (±1.84) |
| O_out | 19.86 (±1.09) | 5.09 ● (±2.41) | 10.61 ● (±5.26) | 9.94 ● (±6.23) | 11.82 (±2.46) | 11.92 (±3.09) | 12.70 (±4.38) |
| C_in | 19.35 (±0.45) | 7.72 ● (±2.63) | 13.75 ● (±4.16) | 10.87 ● (±5.15) | 14.80 ● (±3.07) | 16.07 (±2.86) | 16.29 (±3.35) |
| C_out | 19.78 (±0.25) | 7.09 ● (±3.21) | 13.37 ● (±4.42) | 12.59 ● (±3.80) | 15.34 (±3.22) | 15.84 (±3.10) | 16.55 (±2.09) |
| S_in | 18.32 (±1.18) | 6.25 ● (±3.42) | 12.93 ● (±2.72) | 10.67 ● (±1.85) | 12.78 ● (±3.46) | 13.86 (±2.95) | 14.92 (±4.48) |
| S_out | 19.01 (±1.04) | 5.45 ● (±2.75) | 9.69 ● (±6.27) | 13.80 ● (±3.15) | 12.62 ● (±2.41) | 15.31 (±2.13) | 15.88 (±3.72) |
| N_in | 18.48 (±1.25) | 4.29 ● (±2.22) | 13.85 ● (2.83) | 13.69 ● (±2.21) | 10.96 ● (±3.27) | 13.51 ● (±3.07) | **15.57** (±2.95) |
| N_out | 18.26 (±1.11) | 5.19 ● (±2.47) | 11.83 ● (±3.82) | 14.07 ● (±2.56) | 12.75 ● (±3.18) | 14.29 ● (±3.10) | **16.38** (±2.72) |

Table A10: Average final scores on modified Pong (POMDP) with $N_{targets} = 20$. The best non-oracle results are marked in red, while **bold** indicates a statistically significant result w.r.t. all the baselines. O, C, S, and N denote the orientation, color, size, and noise factors, respectively.

|  | Oracle Upper bound | Non-t lower bound | PNN (Rusu et al., 2016) | PSM (Agarwal et al., 2021a) | MTQ (Fakoor et al., 2020) | AdaRL* Ours w/o masks | AdaRL Ours |
|---|---|---|---|---|---|---|---|
| O_in | 18.65 (±2.43) | 8.04 ● (±1.78) | 12.19 ● (±3.07) | 12.37 ● (±2.92) | 14.64 ● (±3.01) | 17.42 ● (±2.20) | **18.85** (±1.63) |
| O_out | 19.86 (±1.09) | 6.97 ● (±1.88) | 16.48 (±3.10) | 15.79 ● (±2.29) | 12.75 ● (±4.93) | 17.25 (±1.85) | 17.93 (±2.41) |
| C_in | 19.35 (±0.45) | 8.09 ● (±3.11) | 15.89 ● (±3.49) | 16.70 ● (±2.38) | 17.85 (±2.16) | 17.73 ● (±2.01) | 18.93 (±1.37) |
| C_out | 19.78 (±0.25) | 7.48 ● (±2.09) | 16.85 ● (±3.17) | 16.29 ● (±2.64) | 17.93 ● (±2.35) | 18.49 (±2.04) | 19.28 (±1.36) |
| S_in | 18.32 (±1.18) | 7.45 ● (±3.15) | 12.89 ● (±2.04) | 13.84 ● (±3.27) | 15.33 ● (±2.03) | 15.79 ● (±2.62) | **17.49** (±2.18) |
| S_out | 19.01 (±1.04) | 7.04 ● (±2.36) | 14.69 ● (±2.03) | 17.25 ● (±2.30) | 18.48 (±1.36) | 17.82 ● (±1.98) | 19.21 (±0.63) |
| N_in | 18.48 (±1.25) | 6.82 ● (±2.09) | 13.84 ● (±2.82) | 16.80 ● (±1.73) | 17.58 (±2.19) | 15.93 ● (±3.68) | 18.25 (±1.81) |
| N_out | 18.26 (±1.11) | 7.82 ● (±2.46) | 14.89 ● (±2.98) | 16.85 (±3.94) | 17.03 (±2.36) | 16.49 ● (±3.25) | 17.85 (±2.16) |

Table A11: Average final scores on modified Pong (POMDP) with $N_{targets} = 10000$. The best non-oracle results are marked in red, while **bold** indicates a statistically significant result w.r.t. all the baselines. O, C, S, and N denote the orientation, color, size, and noise factors, respectively.

|  | Oracle Upper bound | Non-t lower bound | PNN (Rusu et al., 2016) | PSM (Agarwal et al., 2021a) | MTQ (Fakoor et al., 2020) | AdaRL* Ours w/o masks | AdaRL Ours |
|---|---|---|---|---|---|---|---|
| Rl_in | 7.98 (±3.81) | 3.19 ● (±2.27) | 4.95 (±1.08) | 5.04 (±2.11) | 4.78 (±2.10) | 3.49 ● (±1.97) | 5.81 (±2.06) |
| Rl_out | 9.61 (±4.78) | 5.19 ● (±2.80) | 5.89 (±1.93) | 6.03 (±2.71) | 6.21 (±3.14) | 5.64 (±2.59) | 6.12 (±3.45) |
| Rn_in | 7.62 (±2.16) | 2.85 ● (±1.71) | 5.31 (±2.78) | 5.06 (±3.89) | 5.52 (±3.47) | 5.79 (±3.03) | 5.84 (±3.17) |
| Rn_out | 41.36 (±5.70) | 21.73 ● (±8.54) | 27.19 (±5.82) | 23.27 ● (±8.01) | 25.49 ● (±6.18) | 26.33 ● (±7.94) | 29.92 (±6.39) |

Table A12: Results on modified Pong game with $N_{targets} = 50$. The best non-oracle results are marked in red, while **bold** indicates a statistically significant result w.r.t. all the baselines. Rl and Ro denote the linear reward and nonlinear reward-changing cases, respectively.

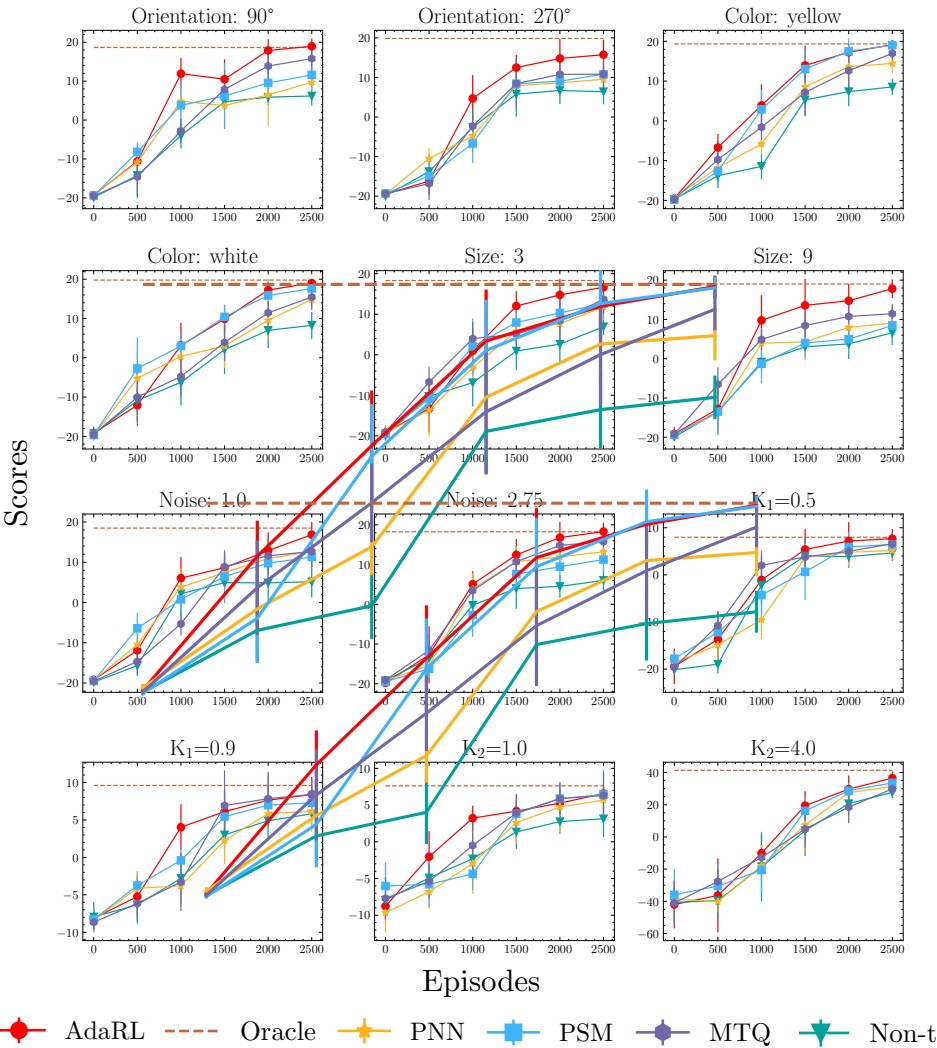

Figure A12: Learning curves for modified Pong experiments with change factors. The reported scores are averaged across 30 trials.

| | Oracle
Upper bound | Non-t
lower bound | PNN
(Rusu et al., 2016) | PSM
(Agarwal et al., 2021a) | MTQ
(Fakoor et al., 2020) | AdaRL*
Ours w/o masks | AdaRL
Ours |
|---|---|---|---|---|---|---|---|
| Rl_in | 7.98
(±3.81) | 4.65 ●
(±1.70) | 5.17 ●
(±1.98) | 6.45 ●
(±1.82) | 6.62 ●
(±2.45) | 6.88 ●
(±3.19) | **7.69**
(±2.04) |
| Rl_out | 9.61
(±4.78) | 5.82 ●
(±2.01) | 6.15 ●
(±2.79) | 7.30 ●
(±1.98) | 8.42
(±2.14) | 7.04 ●
(±2.52) | 8.41
(±2.36) |
| Rn_in | 7.62
(±2.16) | 3.13 ●
(±2.47) | 5.68 ●
(±1.42) | 6.42
(±3.31) | 6.30
(±3.19) | 5.52 ●
(±1.09) | 6.57
(±1.24) |
| Rn_out | 41.36
(±5.70) | 27.70 ●
(±3.45) | 31.28 ●
(±4.09) | 33.60 ●
(±5.52) | 29.77 ●
(±3.85) | 33.83 ●
(±5.02) | **36.52**
(±4.18) |

Table A13: Average final scores on modified Pong (POMDP) with $N_{target} = 10000$. The best non-oracle results are marked in red, while **bold** indicates a statistically significant result w.r.t. all the baselines. Rl and Ro denote the linear and nonlinear reward changes, respectively.

the target domains are specified in Table A15, suggesting that AdaRL can achieve better performance than the meta-learning approaches (i.e., MAML and CAVIA).

|        | Oracle
Upper bound | Non-t
lower bound | PNN
(Rusu et al., 2016) | PSM
(Agarwal et al., 2021a) | MTQ
(Fakoor et al., 2020) | AdaRL*
Ours w/o masks | AdaRL
Ours |
|--------|-----------------------|----------------------|---------------------------|-------------------------------|-----------------------------|-------------------------|---------------|
| Rl_in  | 7.98
($\pm3.81$)   | 4.81 ●
($\pm2.03$) | 5.94 ●
($\pm4.87$)     | 6.90
($\pm3.47$)           | 7.34
($\pm3.18$)         | 6.46 ●
($\pm3.12$)   | 7.93
($\pm2.09$) |
| Rl_out | 9.61
($\pm4.78$)   | 3.89 ●
($\pm2.16$) | 7.85
($\pm2.88$)       | 7.37
($\pm3.75$)           | 8.77
($\pm2.61$)         | 7.78
($\pm3.10$)     | 8.94
($\pm2.02$) |
| Rn_in  | 7.62
($\pm2.16$)   | 3.58 ●
($\pm1.09$) | 6.91
($\pm2.85$)       | 7.01
($\pm2.46$)           | 6.28 ●
($\pm3.14$)       | 6.30 ●
($\pm2.63$)   | 7.57
($\pm1.94$) |
| Rn_out | 41.36
($\pm5.70$)  | 29.98 ●
($\pm3.02$)| 36.08 ●
($\pm10.35$)   | 37.26
($\pm11.25$)         | 38.48
($\pm12.59$)       | 34.19 ●
($\pm9.36$)  | 41.25
($\pm6.92$) |

Table A14: Results on modified Pong game with $N_{targets} = 50000$. The best non-oracle results are marked in red, while **bold** indicates a statistically significant result w.r.t. all the baselines. Rl and Ro denote the linear reward and nonlinear reward-changing cases, respectively.

|              | MAML
(Finn et al., 2017) | CAVIA
(Zintgraf et al., 2019) | AdaRL
Ours |
|--------------|-----------------------------|----------------------------------|---------------|
| Cheetah (vel) | $-89.8$
($\pm4.1$)       | $-86.5$
($\pm2.0$)            | **-81.7**
($\pm3.2$) |
| Ant (vel)     | 100.4
($\pm10.9$)        | 95.7
($\pm6.92$)              | **106.8**
($\pm8.4$) |

Table A15: Results on MuJoCo benchmarks (Cheetah and Ant experiments with different target velocities, with 30 trials each) with $N_{targets} = 50,000$. The best results are marked in **red**.

### A5.5 EFFECT OF THE POLICY USED FOR DATA COLLECTION DURING MODEL ESTIMATION

In our framework, we use the random policy to generate trajectories for each domain. The generated trajectories are further used to conduct the model estimation. To study whether the random policy will affect the effectiveness of model estimation, we compare the learnt parameters of latent space in model estimation with the trajectories generated via (1) the random policy in our framework, and (2) the optimal policies learnt on the source domains. Here we show the case with changing gravity in the modified Cartpole game. In Fig. A13, we give the learnt parameters ($\mu$ and $\log \sigma$) of the first and second components in the Gaussian mixtures of the 10-th latent state at 20-th epoch. The results demonstrate that the difference between the two sets of learnt parameters is limited.

### A5.6 MORE STATISTICAL EVALUATION PROTOCOLS ON THE PERFORMANCE

In this section, we provide a more detailed and comprehensive comparison on the performances of AdaRL and other baseline methods. Following the recent published work on reliable evaluation for RL (Agarwal et al., 2021b), we utilize 4 evaluation metrics: median, interquartile mean (IQM), mean, and optimality gap. Median and mean are the sample median and mean, respectively. IQM discards the top and bottom 25% samples and then computes the mean value of the remaining ones. Thus, this factor is insensitive to outliers. Optimality gap is quantified via the gap between the performance of each method and the mean score obtained by the oracle agent. Therefore, higher mean, median and IQM and lower optimality are the indications of better methods. All metrics are with 95% bootstrap confidence intervals (CIs) (Efron, 1992).

Fig. A14, A15, A16, and A17 give the comparison on AdaRL and baselines using the set of evaluation protocols in different games and settings. The results suggest that in most cases, AdaRL performs consistently better than the baselines across all evaluation metrics.

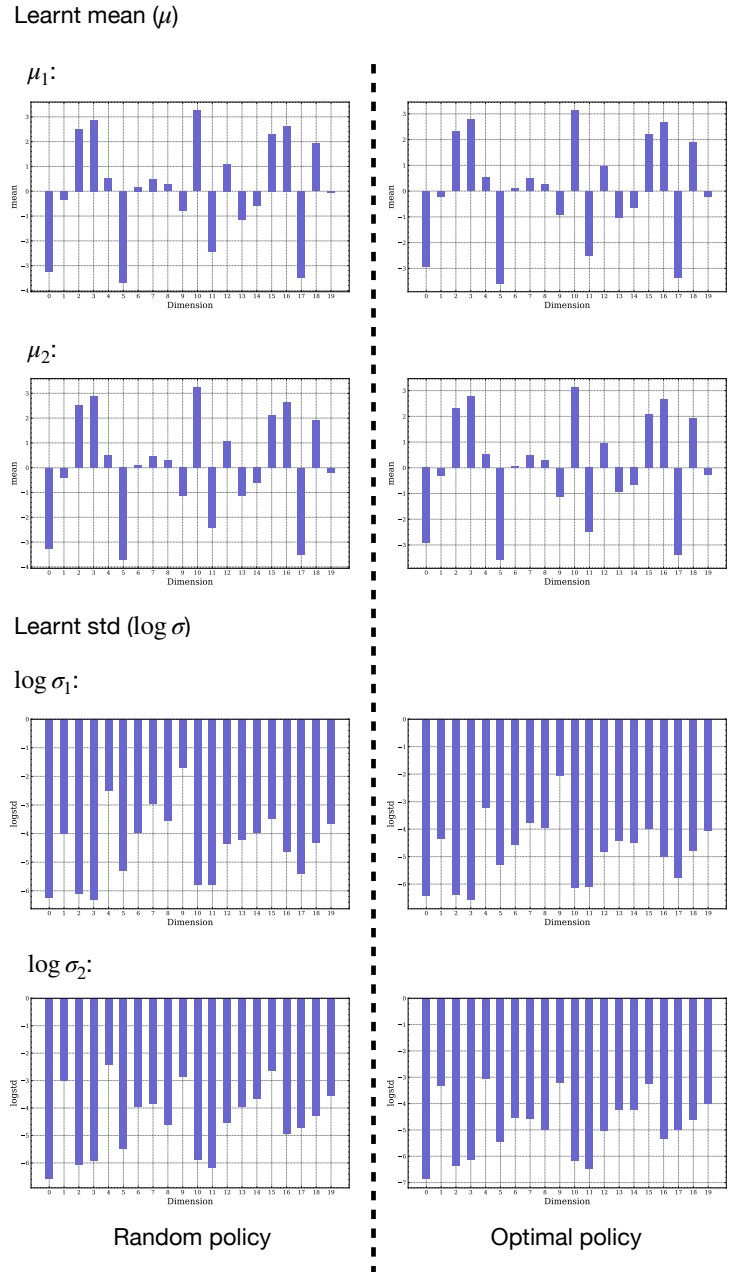

Figure A13: Learnt $\mu$ and $\log \sigma$ in model estimation with both random policy (left), and optimal policy (right).

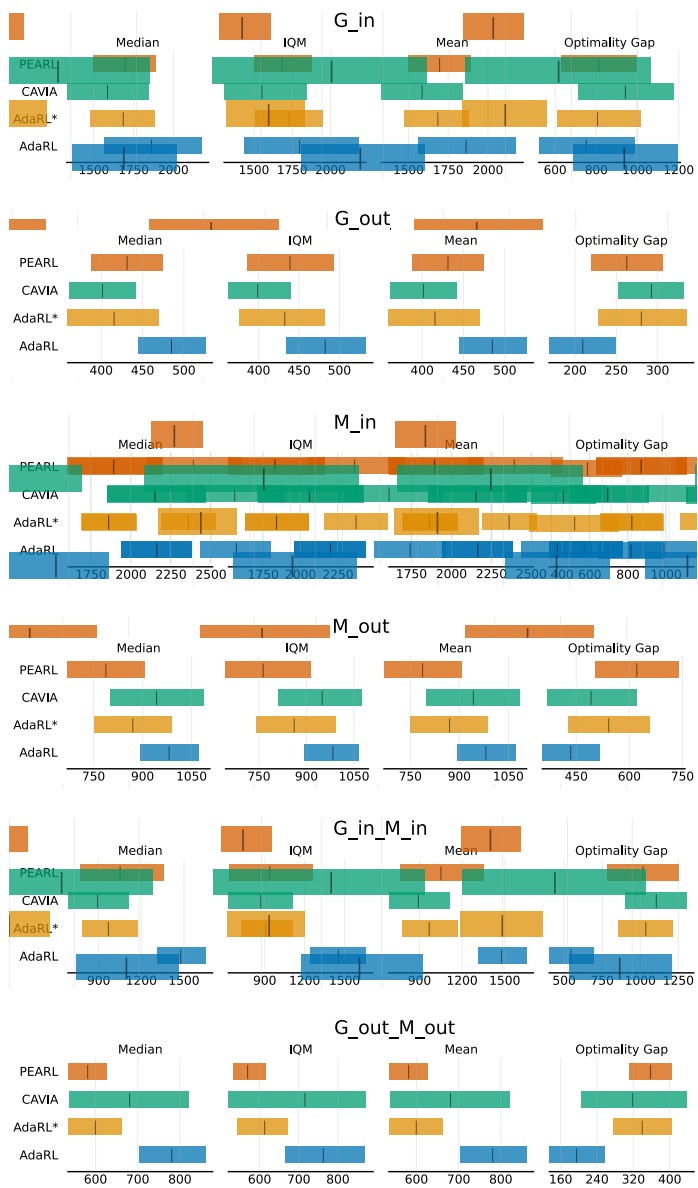

Figure A14: Evaluation on the results of modified CartPole game under MDP settings for $N_{target} = 50$.

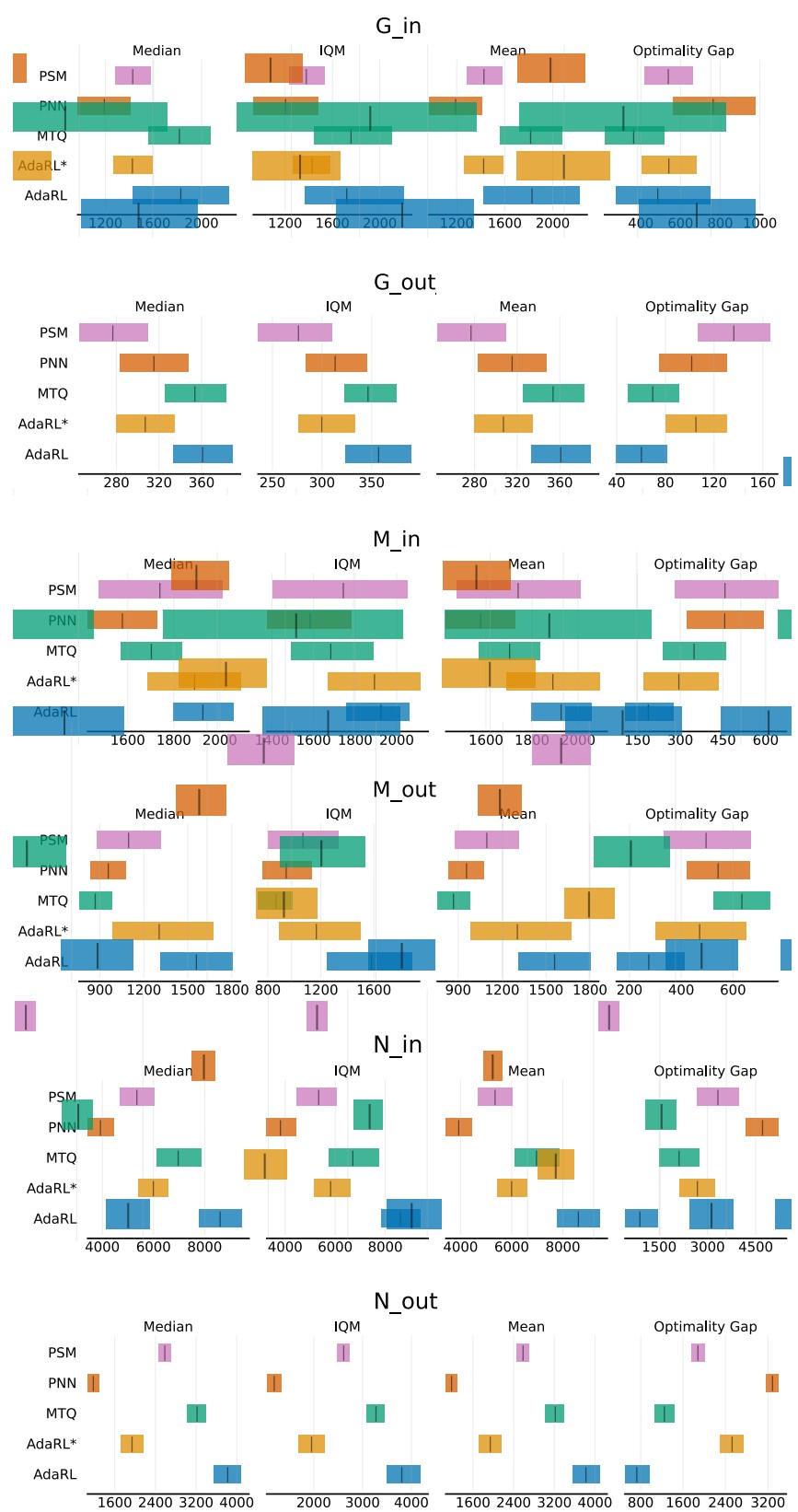

Figure A15: Evaluation on the results of modified CartPole game under POMDP settings for $N_{target} = 50$.

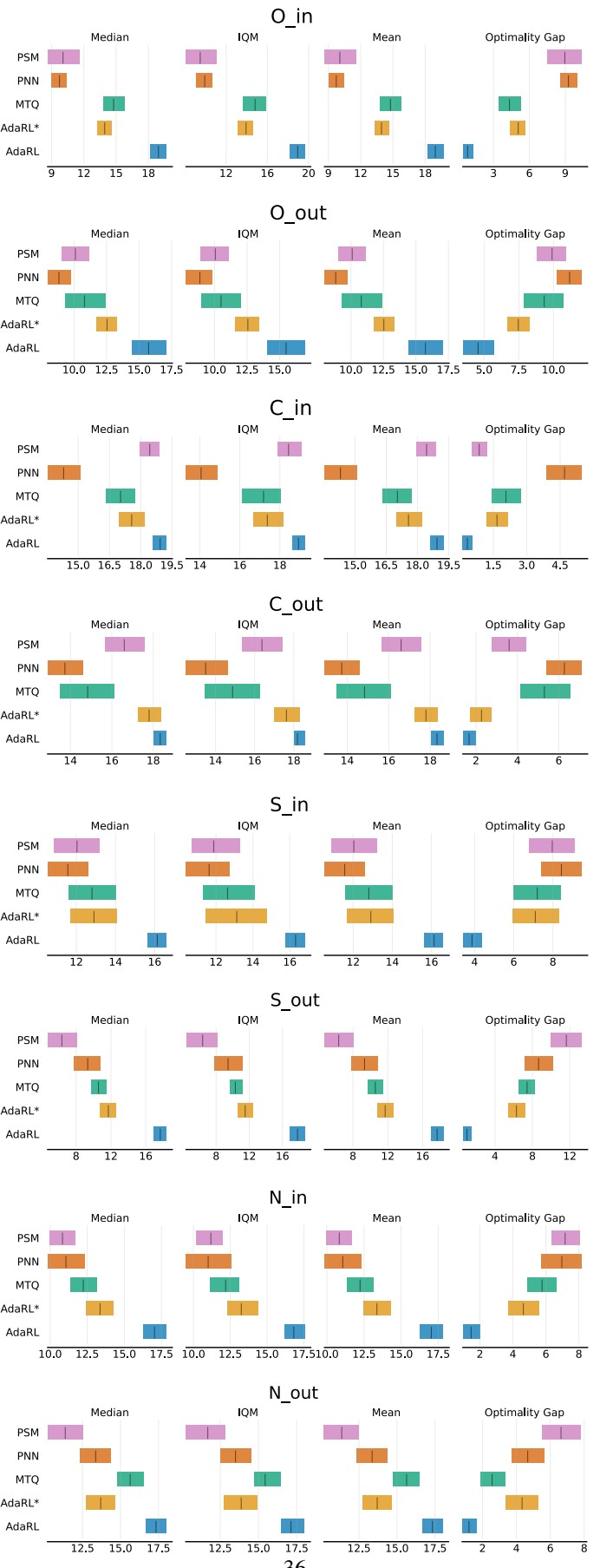

Figure A16: Evaluation on the results of modified Atari Pong game under POMDP settings with changing orientation, color, size, and noise levels for $N_{target} = 50$.

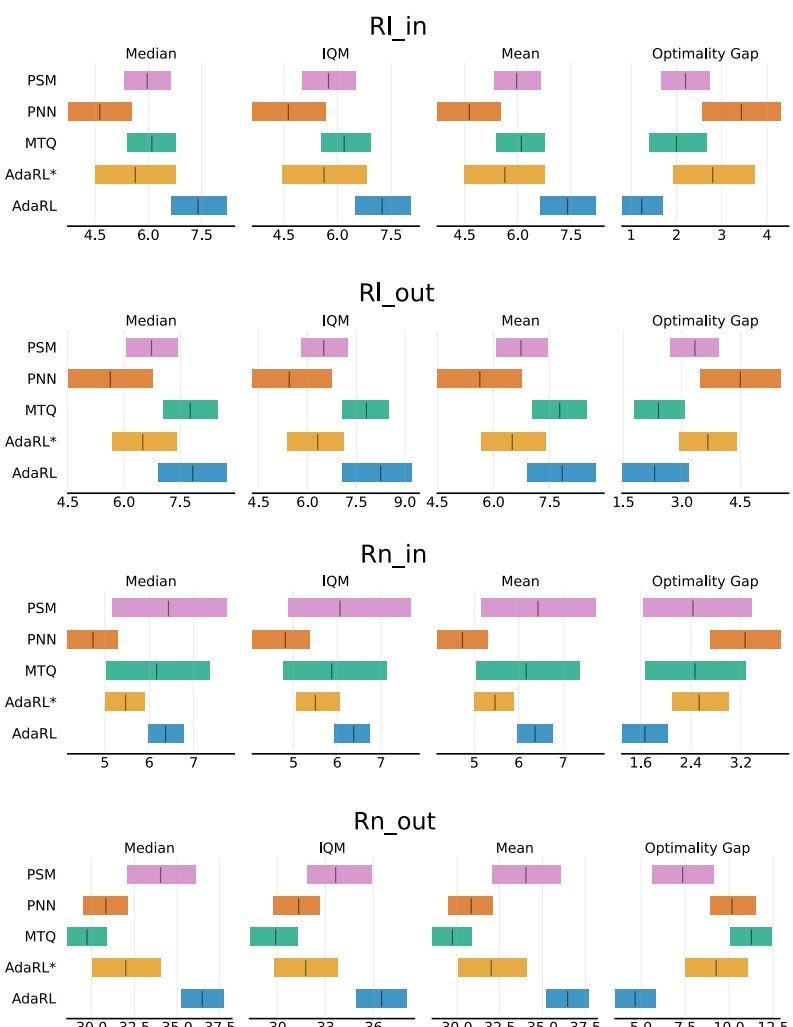

Figure A17: Evaluation on the results of modified Atari Pong game under POMDP settings with changing reward functions for $N_{target} = 10000$.

## A5.7 Pseudo code of Miss-VAE

**Algorithm A1** Pseudo code of Miss-VAE.

```
1   # s: latent state; theta_o, theta_s, theta_r: changing factors;
2   # C_so, C_sr, C_ar, C_ss, C_as, C_theta_oo, C_theta_rr, C_theta_ss: structure masks;
3   # o: observation; a: action; r: reward;
4   # num_mix: the number of components in mixture Gaussian model
5
6   ####################################### Encoder #######################################
7   for X in loader: # load a batch of data from K domains
8       ho = conv(o)
9       ha = conv(a)
10      hr = conv(r)
11      hd = fc(theta_o, theta_s, theta_r)
12      h = concat[ho, ha, hr, hd]
13      output, last_state = LSTM(h) # output the logmix, mean, and logstd of state
14      s = sample(output) # sample the state
15
16  ####################################### Decoder #######################################
17      # Reconstruct o_{t}
18      s_o = multiply(s, C_so)
19      theta_o_o = multiply(theta_o, C_theta_oo)
20      h_s_theta = concat(fc(s_o), fc(theta_o_o))
21      o_t = conv_transpose(h_s_theta) # observation of current step
22
23      # Reconstruct r_{t+1}
24      s_r = multiply(s, C_sr)
25      a_r = multiply(a, C_ar)
26      theta_r_r = multiply(theta_r, C_theta_rr)
27      h_as = fc(concat[s_r, a_r])
28      h_theta_r = fc(theta_r_r)
29      r_t+1 = fc(concat[h_as, h_theta_r]) # reward of current step
30
31      # Predict O_{t+1}
32      o_t+1 = conv_transpose(s, theta_o, theta_s) # reward of next step
33
34      # Predict r_{t+2}
35      r_t+2 = fc(concat[fc(s), fc(a), fc(theta_o), fc(theta_s)]) # reward in the next step
36
37      # Markovian transition
38      s_s = multiply(s, C_ss)
39      theta_s_s = multiply(theta_s, C_theta_ss)
40      a_s = multiply(a, C_as)
41      h_dyn = fc(concat[s_s, theta_s_s, a_s])
42      s_output = linear(h_dyn)
43      s_logmix, s_mean, log_logstd = s_output.split # state in next step
44
45  ####################################### Loss #######################################
46      # Reconstruction loss
47      Rec_loss = mean(MSE(o - o_t) + MSE(r - r_t+1))
48
49      # Prediction loss
50      Pred_loss = mean(MSE(o[:,:-1,:] - o_t+1) + MSE(r[:,:-1,:] - r_t+2))
51
52      # KL loss
53      for idx in range(num_mixture):
54          g_logmix, g_mean, g_logstd = s_logmix[:,idx], s_mean[:,idx], log_logstd[:,idx]
55          KL_Loss += KL(g_logmix, g_mean, g_logstd)
56      KL_loss = mean(log(1 / (KL_Loss + 1e-10) + 1e-10))
57
58      # Sparsity constraints
59      Reg_loss = mean(C_so) + mean(C_sr) + mean(C_ar) + mean(C_ss) + mean(C_theta_oo) +
            mean(C_theta_rr) + mean(C_theta_ss) + mean(theta_o) + mean(theta_s) + mean(
            theta_r)
60
61      # Optimization step
62      loss = Rec_loss + Pred_loss + KL_loss + Reg_loss
63      loss.backward()
64      optimizer.step()
```

## A6 Experimental details

### A6.1 Hyperparameters selections

**Model estimation**    We use a random policy to collect sequence data from source domains. For both modified Cartpole and Pong experiments, the sequence length is 40 and the number of sequence is 10000 for each domain. The sampling resolution is set to be 0.02. Other details are summarized in Table A16. Parts of the implementation of model estimation modules follows the open-source framework in the world model (Ha and Schmidhuber, 2018).

| Settings | Cartpole | Pong |
|---|---|---|
| # Dimensions of latent space | 20 | 25 |
| # Dimensions of $\theta$ | 1 | Size & noise: 1, orientation: 2, color: 3, Reward: 1 (linear), 2 (non-linear) |
| # Epochs | 1000 | reward-varying: 4000, others: 1500 |
| Batch size | 20 | 80 |
| # RNN cells | 256 | 256 |
| Initial learning rate | 0.01 | 0.01 |
| Learning rate decay rate | 0.999 | 0.999 |
| Dropout | 0.90 | 0.90 |
| KL-tolerance | 0.50 | 0.50 |

Table A16: Experimental details on the model estimation part.

**Policy learning**    We adopt Double DQN (Van Hasselt et al., 2016) during policy learning stage. The detailed hyper-parameters are summarized in Table A17. For a fair comparison, we use the same set of hyperparameters for training other baseline methods.

| Settings | Cartpole | Pong |
|---|---|---|
| Discount factor | 0.99 | 0.99 |
| Exploration rate | 1.0 | 1.0 |
| Initial learning rate | 0.01 | 0.01 |
| Learning rate decay rate | 0.999 | 0.999 |
| Dropout | 0.10 | 0.10 |

Table A17: Experimental details on the policy learning part.

### A6.2 Experimental platforms

For the model estimation, Cartpole and Pong experiments are implemented on 1 NVIDIA P100 GPUs and 4 NVidia V100 GPUs, respectively. The policy learning stages in both experiments are implemented on 8 Nvidia RTX 1080Ti GPUs.

### A6.3 Licenses

In our code, we have used the following libraries which are covered by the corresponding licenses:

- Tensorflow (Apache License 2.0),
- Pytorch (BSD 3-Clause "New" or "Revised" License),
- OpenAI Gym (MIT License),
- OpenCV (Apache 2 License),
- Numpy (BSD 3-Clause "New" or "Revised" License)
- Keras (Apache License).

We released our code under the MIT License.

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
