# OpenReview forum: "AdaRL: What, Where, and How to Adapt in Transfer Reinforcement Learning"
_ICLR.cc/2022/Conference — ICLR 2022 Spotlight_

### Official Review · Reviewer_i8j9 · 2021-11-02

**Correctness:** 4
**Technical Novelty And Significance:** 3
**Empirical Novelty And Significance:** 3
**Recommendation:** 8
**Confidence:** 4

**Main Review:**

The paper investigates the question of transfer via identifying a small number of changing variables to maximise data efficiency. It is overall clearly written and easy to follow for most parts.

The three step training framework with a split between model estimation from random data and policy learning can lead to a set of challenges which should be mentioned and discussed. First, random data might not cover enough of an environment's state space such that the estimated model is not accurate for later stages of policy training when a different part of the state space is visited. Second, splitting model estimation (focused on reconstruction and prediction) and policy training can lead to model features which are suboptimal for the policy. Third, the policy will be trained with the feature distributions of a set of source domains. Any function approximator used for the policy will need to handle out of distribution data on target domains which can differ from these features. In particular the second point could be investigated by iterating between model estimation and policy training.

The evaluation focuses on new variations of known domains opening the question why no existing benchmarks e.g. from the metalearning literature are used (e.g. benchmarks from the set of baseline papers such as the cheetah or ant domains from MAML). Further, it is unclear why e.g. MAML, Progressive Networks, etc (which are used in the MDP setting) are not applied to the POMDP setting. Vice versa for many POMDP methods which are not constrained to the use of POMDPs.

The use of image based domains does not need to lead to a POMDP. In both cartpole and Pong, occlusions do not occur. If the agent stacks multiple frames, velocity as part of the state is included. If not, it will be helpful to emphasise this aspect to describe which aspects are unobserved.

While overall well written, parts about equations and the general model are hard to trace and a visual representation of various parts would be helpful. See minor points for details.

Minor:
- description of meaning for masking parameters (c’s) should be move closer to Eq. 1
- a graphical representation for Eq. 1 (definition 1 with respect to compactness would benefit from an example graph either in the appendix or, if space can be found, in the main paper)
- Section 3.1. describes the training of the VAE via SGD. Including ‘in one step’ in the title seems inaccurate here. I could not find any justification for this term in the section.
- Explicit pseudocode for algorithm 1 is very helpful!
- The related work section provides a broad background but work on learning an embedding space for system dynamics to enable quick adaptation for sim2real transfer should be added [1,2].


[1] Yu, Wenhao, et al. "Preparing for the unknown: Learning a universal policy with online system identification." arXiv preprint arXiv:1702.02453 (2017).
[2] Peng, Xue Bin, et al. "Learning agile robotic locomotion skills by imitating animals." arXiv preprint arXiv:2004.00784 (2020).


**Summary Of The Paper:**

The submission proposes a method for transfer in reinforcement learning building on estimating a small set of factors describing a system and modelling the agent as a dynamic bayesian network. In detail, the method splits into domain shared and domain dependent factors which are modelled via a new form of structured sequential VAE. The VAE is trained to enable dynamics, observation and reward prediction (including reconstruction, prediction, KL and sparsity regularisation losses).

Data for VAE training is collected from random policies and model fitting is followed by policy training on source domains to be transferred to target domains by identifying domain specific features for the target domains.

The method is evaluated on variations of a cartpole and a pong domain (both from images) and evaluated against a set of recent, competitive baselines.


**Summary Of The Review:**

A strong submission on improving transfer learning which leaves a couple of open questions regarding its evaluation. However, these questions should be easy to address in the rebuttal.

---

> ### Author Response · Authors · 2021-11-19
> **Thanks for the review - some additional information (part 1)**
>
> We appreciate your time dedicated to reviewing this paper and your thoughtful comments and suggestions.
>
> > Q1: The three step training framework with a split between model estimation from random data and policy learning can lead to a set of challenges which should be mentioned and discussed. Random data might not cover enough of an environment's state space such that the estimated model is not accurate for later stages of policy training when a different part of the state space is visited. Splitting model estimation (focused on reconstruction and prediction) and policy training can lead to model features which are suboptimal for the policy. In particular this could be investigated by iterating between model estimation and policy training.
>
> Thank you for your thoughtful comments and suggestions. We will including corresponding discussions in the paper once all relevant experiments are complete.
>
> We agree that random data might not cover enough of the environment's state space. So, to mitigate this issue, we randomly and uniformly chose the initial state. Here we show a preliminary comparison of our original result with the new results in which we use the learned optimal policies to generate the data for model estimation.
>
> Specifically, for the MDP case, with the learned optimal policy to generate the data for model estimation, the final averaged score across $30$ trials is $2294.8$, while the original score with the random behavior policy is $2217.6$ (see Table 1 in the paper). This demonstrates that using optimal policies indeed improves the model estimation and further policy optimization.
>
> For the POMDP case, we compare the estimated model parameters with the data generated from (a) random policy exploited in our framework and that from (b) the optimal policy learnt on the source domains. The comparison is given in Fig. A13 (Section A5.5) in the updated appendix.
>
> One can see that the differences between the two sets of learned parameters are small, suggesting that our adopted random policy can also lead to good model estimation.
>
> Furthermore, we appreciate your suggestion on iterating between model estimation and policy training. We have done an additional experiment with the following steps:
>
> 1) Estimate the model in Eq. 1 using the rollouts generated by random policy. The number of rollouts is $6000$ per domain.
>
> 2) Conduct policy learning and model estimation iteratively. Specifically, we update the model estimation every $50$ epochs, and we only leverage the latest trajectories collected by the optimized policies to update the model. The total number of the episodes is $500$.
>
> The following table reports the results in the setting of changing gravity in Cartpole (MDP); the first two columns give the results from 2 trials with 1 iteration (we are waiting for the results of more trials), and the 3rd column gives the original results (mean and standard deviation) without iterations. We are running more experiments to estimate the mean and standard deviation of the scores with iterations:
>
> |       | Iterative AdaRL Trial 1 | Iterative AdaRL Trial 2 |      AdaRL     |
> |:-----:|:---------------------:|:---------------------:|:--------------:|
> |  G_in |          2414         |          2975         | 2217.6 (981.5) |
> | G_out |          556          |          583          |  508.3 (138.2) |
>
>
> > Q2: Third, the policy will be trained with the feature distributions of a set of source domains. Any function approximator used for the policy will need to handle out of distribution data on target domains which can differ from these features.
>
> Thanks for raising this point. Similar to the task of domain adaptation, if the data distribution is not in the support of that of the source domains, the current solution may not be optimal. How to appropriately handle this case will be our future work.

---

> ### Author Response · Authors · 2021-11-19
> **Thanks for the review - some additional information (part 2)**
>
> > Q3: The evaluation focuses on new variations of known domains opening the question why no existing benchmarks e.g. from the metalearning literature are used (e.g. benchmarks from the set of baseline papers such as the cheetah or ant domains from MAML). Further, it is unclear why e.g. MAML, Progressive Networks, etc (which are used in the MDP setting) are not applied to the POMDP setting. Vice versa for many POMDP methods which are not constrained to the use of POMDPs.
>
> Thanks for the suggestions, we have tried to cover all the reviewer's concerns.
>
> We have conducted a series of additional experiments on other benchmarks. Specifically, following MAML and CAVIA, we have added experiments on MuJoCo benchmarks (Todorov et al., 2012), including Cheetah and ant with different target velocities. The average final scores (including mean and standard deviation across 30 trials) are summarized below, which suggest that AdaRL achieves better performance than the meta-learning approaches (i.e., MAML and CAVIA); see more details in Section A5.4 in the updated appendix. We will further update the manuscript once we have the complete experimental results.
>
> | | MAML | CAVIA| AdaRL|
> |---|-------| ---- | -----|
> |Cheetah (vel)| -89.8 +/- 4.1 | -86.5 +/- 2.0 | -81.7 +/- 3.2|
> | Ant (vel) | 100.4 +/- 10.9 | 95.7 +/- 6.92 | 106.8 +/- 8.4
>
>
> We have also applied 1) PNN and PSM to the MDP settings, and 2) CAVIA and PEARL to POMDP settings, where we extracted the latent states with the world model (Ha & Schmidhuber, 2018) for meta-updating and policy optimization.
>
> Below, we show a part of the results, which show that AdaRL outperforms all other non-oracle methods consistently. We will update the paper to include full results once they are available.
>
> 1) Apply PNN and PSM to MDP settings with changing gravity in the modified Cartpole game (mean/std are computed across $30$ trials):
>
> |       | Oracle         | Non-t          | PNN            | PSM            | AdaRL*         | AdaRL          |
> |-------|----------------|----------------|----------------|----------------|----------------|----------------|
> | G\_in  | 2486.1 (369.7) | 1098.5 (472.1) | 1349.0 (478.5) | 1745.9 (393.6) | 1940.5 (841.7) | 2217.6 (981.5) |
> | G\_out | 693.9 (100.6)  | 204.6 (39.8)   | 426.8 (97.9)   | 368.2 (146.3)  | 439.5 (157.8)  | 508.3 (138.2)  |
>
>
> 2) Apply CAVIA and PEARL to POMDP settings with changing noise levels in the modified Pong game (mean/std are computed across $30$ trials):
>
> |       | Oracle       | Non-t       | CAVIA        | PEARL        | AdaRL*       | AdaRL        |
> |-------|--------------|-------------|--------------|--------------|--------------|--------------|
> | N\_in  | 18.48 (1.25) | 5.51 (3.88) | 13.96 (3.27) | 13.05 (4.13) | 13.78 (2.15) | 16.84 (3.13) |
> | N\_out | 18.26 (1.11) | 6.02 (3.19) | 12.79 (3.58) | 13.49 (2.83) | 14.65 (3.01) | 18.30 (2.24) |
>
> > Q4: The use of image based domains does not need to lead to a POMDP. In both cartpole and Pong, occlusions do not occur. If the agent stacks multiple frames, velocity as part of the state is included. If not, it will be helpful to emphasise this aspect to describe which aspects are unobserved.
>
> We did not stack multiple frames, so some properties (e.g., cart velocity and pole angular velocity in Cartpole) are not observed. We have made it clear in the first paragraph of Section 4 of the updated manuscript. Moreover, even if we stacked the frames, there are still other properties (e.g., mass of the pole in Cartpole) that may not be observed, leading to a POMDP.
>
>
>
> > Q5: Parts about equations and the general model are hard to trace and a visual representation of various parts would be helpful.
>
> Thanks for the suggestion. Please refer to Figures A1 and A2 in Appendix for illustrations and explanations of Eq. 1, which were omitted in the main text due to a lack of space.
>
> > Q6: description of meaning for masking parameters (c’s) should be move closer to Eq. 1
>
> The masking parameters $\mathbf{c}$ were briefly described in the second line right after Eq. (1). Then in the follow-up paragraph, we gave a detailed description of the structural relationships and graphs.

---

> ### Author Response · Authors · 2021-11-19
> **Thanks for the review - some additional information (part 3)**
>
> > Q7: a graphical representation for Eq. 1 (definition 1 with respect to compactness would benefit from an example graph either in the appendix or, if space can be found, in the main paper)
>
> Please see the graphical representation of Eq 1 in Figure A1 and several examples given in Figure A2 in Appendix.
>
> > Q8: Section 3.1. describes the training of the VAE via SGD. Including ‘in one step’ in the title seems inaccurate here. I could not find any justification for this term in the section.
>
> We have replaced "in one step" with "simultaneous", as a way to emphasize that models from different domains are estimated together simultaneously, instead of estimating the model in each domain separately.
>
> > Q9: Explicit pseudocode for algorithm 1 is very helpful!
>
> We have added a Python-like pseudocode of model estimation in Section A5.7 in the updated appendix.
>
> > Q10: The related work section provides a broad background but work on learning an embedding space for system dynamics to enable quick adaptation for sim2real transfer should be added [1,2].
>
> Thank you for pointing us to these related works. We have added them to the updated version.

---

> > ### Comment · Reviewer_i8j9 · 2021-11-24
> > **Feedback**
> >
> > Thank you very much for your detailed feedback. In particular the clarification around POMDP creation, rephrasing of paragraphs and additional experiments for meta learning are highly appreciated. The additional information and experiments further supports the current rating, a strong submission.
> >
> > On a side note: thanks for adding additional pseudocode on a part of the algorithm.
> > The original Q9 was actually not planned as a request but rather a comment on a helpful part of the submission!

---

### Official Review · Reviewer_6qyY · 2021-11-02

**Correctness:** 4
**Technical Novelty And Significance:** 3
**Empirical Novelty And Significance:** 2
**Recommendation:** 8
**Confidence:** 4

**Main Review:**

Overall I quite liked this paper as it introduces a well-motivated idea that can be quite useful to the research community. However, the proposed algorithm is rather involved and the empirical gains seem to be quite marginal relative to other methods in the literature (which are arguably easier to explain/implement).

My two main concerns are clarity and the quality of the empirical evaluations. More details below.

**Clarity**: The idea is somewhat involved and there are _lots_ of parameters and moving parts. While the authors have made a good effort to make it clear, there are still some parts that are hard to follow. Some more details below.

1.  Although Figure 1 helps somewhat, it is still rather difficult to follow. I would suggest moving Figure A3 from the appendix to the main paper.
1.  In the definitions of $\mathcal{L}^{rec}$ and $\mathcal{L}^{pred}$ the $\beta$ terms need to be explained more. Both $\beta_1$ and $\beta_2$ produce $o$s and $r$s, yes? Why are they conditioned on different parameters? I _think_ one is the Dynamics and the other is the Decoder model (as illustrated in Figure A3), but it's not clear. Indeed, the sentence immediately after says that "$p_{\beta_1}$ and $p_{\beta_2}$ denote  the generative model with parameters $\beta_1$ and $\beta_2$". The use of the singular "the" suggests that it's the same model with different parameters, which I don't think is the case.
1. In Algorithm 1, are $Q$ and $Q'$ initialized independently or in the same way? Does it matter?
1. The rightmost panel in Figure 2 is difficult to understand. Maybe show more context of the Pong game so it's clear that this is a paddle and a ball.

**Empirical evaluations:**
1.  Although I can appreciate the environment modifications the authors used to showcase their algorithm, there are already existing benchmarks (such as [ProcGen](https://openai.com/blog/procgen-benchmark/), [Jumping Task](https://github.com/Maluuba/jumping-task), and [Atari modes](https://arxiv.org/abs/1810.00123)). Why were none of these used to evaluate the proposed method?
1.  The bold red fonts in the tables at the end are misleading, as there is _major_ overlap between the AdaRL confidence intervals with many of the other methods, which means that the gains are often not statistically significant. This is also evident in the appendix figures (e.g. Figures A6 and A7). There seem to be no statistically significant gains to be had from AdaRL.

---

Some questions for the authors:
1. In the definition of $\mathcal{L}^{rec}$, what is the range of $q_{\phi}(\cdot |\theta_k)$? $S$? Writing the expectation subscript as $s_{t,k}\sim q_{\phi}(\cdot | \theta_k)$ would help clarify things.
1.  In the last sentence of the **Modified Cartpole setting**, does $\theta_k=\{\theta_k^o\}$ mean the only variations are in observations?
1. In page 8, right above the **Baselines** section, it says $r_t=\frac{kL}{d+3L}$. Is $k$ the actual index used in the computation of $r_t$? If not, use a different letter.

---

Some other minor points:

1. The authors use the word "parsimonious" a lot, which seems strange. The Oxford Dictionary says its definition is _"extremely unwilling to spend money"_, which seems like a strange adjective to apply to an RL algorithm.
1.  In the Introduction the authors should also reference [Contrastive Behavioral Similarity Embeddings for Generalization in Reinforcement Learning](https://arxiv.org/abs/2101.05265). I know it is referenced later in the paper, but it should also be discussed in the introduction when other representation learning methods (such as (Zhang et al. 2021)) are being discussed.
1.  PNN in the introduction is not defined.
1. At the bottom of page two it says "We learn an optimal policy", but to be precise it should be "a near-optimal policy".
1.  In Equation 1 the index $k$ is used. Later in the paper it becomes clear that it indexes domain, but this should be clarified before equation (1).
1.  At the bottom of page 4 the term ${\bf y}_{1:t,k}$ is used, but this variable has not been introduced yet (it is only introduced at the top of page 5).
1. Right above section 3.2 it says "In this way, except $\theta_k$", it's missing a "for" before the $\theta$.
1. In Algorithm 1 (e.g. lines 7 and 13) use `\cdot` in the posterior (e.g. $q(\cdot | o_{\leq t+1, ...})$), as the posterior at this point is a distribution, not a materialized probability.
1. In page 2 of the appendix there's a missing "}" in the first line of the last paragraph.
1. In the first bullet of page 2 of the appendix it should say "then it ***means*** there is no path".

**Summary Of The Paper:**

This paper proposes a method that learns structural relationships between variables of the RL system so as to be able to adapt to changes across domains, and learn a new policy in a new domain with few samples. The authors provide some theoretical results on their method, and evaluate their proposal on some standard environments that have been modified to evaluate this type of idea.

**Summary Of The Review:**

I think this is an interesting paper, but it falls short in the empirical evaluations. I think the idea is interesting and the theoretical results may be of interest to others working in this space.
However, I don't feel the empirical performance justifies the complexity of the algorithm, which is why I'm somewhat borderline, but leaning towards an accept.
I look forward to reading what the other reviewers think, as well as the authors' response.

---

> ### Author Response · Authors · 2021-11-19
> **Thanks for the review - some additional information (part 1)**
>
> We thank the reviewer for the comments and insightful thinking. Below please see our responses as well as clarifications.
>
> > Q1: Although Figure 1 helps somewhat, it is still rather difficult to follow. I would suggest moving Figure A3 from the appendix to the main paper.
>
> A1: Thanks for the suggestion, we did include a more compact version of Figure A3 in the main paper (see Figure 2, Page 5).
>
> > Q2: In the definitions of $\mathcal{L}^{\text {rec }}$ and $\mathcal{L}^{\text {pred }}$ the $\beta$ terms need to be explained more. Both $\beta_{1}$ and $\beta_{2}$ produce $o s$ and $r s$, yes? Why are they conditioned on different parameters? I think one is the Dynamics and the other is the Decoder model (as illustrated in Figure A3), but it's not clear. Indeed, the sentence immediately after says that " $p_{\beta_{1}}$ and $p_{\beta_{2}}$ denote the generative model with parameters $\beta_{1}$ and $\beta_{2}$ ". The use of the singular "the" suggests that it's the same model with different parameters, which I don't think is the case.
>
> A2: Please see below for more explanations of the $\beta$ terms. $\beta_1$ is the set of parameters in the reconstruction component for $\mathcal{L}^{\text{rec}}$, and $\beta_2$ is the set of parameters in the prediction component for $\mathcal{L}^{\text {pred}}$. Please notice that $\mathcal{L}^{\text{rec}}$ and $\mathcal{L}^{\text{rec}}$ are different, where the former one is to **reconstruct** $o_t$ and $r_{t+1}$, and the latter one is to **predict** $o_{t+1}$ and $r_{t+2}$. We have changed that sentence to ``$p_{\beta_{1}}$ and $p_{\beta_{2}}$ denote the generative models with parameters $\beta_{1}$ and $\beta_{2}$, respectively." Thanks for raising the good point, which helps improve the presentation.
>
> > Q3: In Algorithm 1, are  $Q$ and  $Q^{\prime}$ initialized independently or in the same way? Does it matter?
>
> A3: $Q$ and $Q^{\prime}$ are initialized in the same way, as that in Q-learning (Mnih et al., 2015).
>
>
> > Q4: The rightmost panel in Figure 2 is difficult to understand. Maybe show more context of the Pong game so it's clear that this is a paddle and a ball.
>
> A4: Thank you for the comment and suggestion. We have updated it in the revised version accordingly.
>
>
> > Q5: Although I can appreciate the environment modifications the authors used to showcase their algorithm, there are already existing benchmarks (such as ProcGen, Jumping Task, and Atari modes). Why were none of these used to evaluate the proposed method?
>
> A5: Thanks for the suggestion. We are now running our method on the benchmarks you recommended, starting with Atari modes. We will update the results once they are available. We are also testing AdaRL on the MuJoCo locomotion tasks, which involve rather complex dynamics. We have just obtained some preliminary results on two MuJoCo tasks, Cheetah and Ant, with different target velocities, following the setup in MAML and CAVIA. The average final scores (including mean and standard deviation across 30 trials) are summarized below, which suggest that AdaRL achieves better performance than the meta-learning approaches (i.e., MAML and CAVIA); see more details in Section A5.4 in the updated appendix. We will further update the manuscript once we have the complete experimental results.
>
> | | MAML | CAVIA| AdaRL|
> |---|-------| ---- | -----|
> |Cheetah (vel)| -89.8 +/- 4.1 | -86.5 +/- 2.0 | -81.7 +/- 3.2|
> | Ant (vel) | 100.4 +/- 10.9 | 95.7 +/- 6.92 | 106.8 +/- 8.4|

---

> ### Author Response · Authors · 2021-11-19
> **Thanks for the review - some additional information (part 2)**
>
> > Q6: The bold red fonts in the tables at the end are misleading, as there is major overlap between the AdaRL confidence intervals with many of the other methods, which means that the gains are often not statistically significant. This is also evident in the appendix figures (e.g. Figures A6 and A7). There seem to be no statistically significant gains to be had from AdaRL.
>
> Thanks for the suggestion! We have conducted the Wilcoxon signed-rank test to verify the statistical significance of the performance gain of AdaRL compared with the baselines. We have revised Table 1-3 (in main paper) and all the tables in the appendix with the updated results. Specifically, the bullet indicates whether the performance gains of AdaRL are statistically significant w.r.t. the corresponding baseline (via Wilcoxon signed-rank test at $5\%$ significance level), and the bold indicates a statistically significant results w.r.t. all baselines. The results demonstrate that the improvement of AdaRL is statistically significant in most cases, especially those with POMDP settings, which are more challenging.
>
> Furthermore, in addition to the mean and standard deviation of the scores, we have provided a more thorough statistical analysis in Section A5.6 in the updated appendix, following Agarwal et al. (2021). The additional evaluation metrics include median, interquartile mean (IQM), mean, and optimality gap, which are more robust and efficient aggregate metrics (Agarwal et al. 2021); all are with $95\%$ bootstrap confidence intervals. Basically, a higher mean, median, and IQM, and lower optimality are the indications of a better performance. Fig. A14 - A17 in the updated appendix demonstrate that in most cases, AdaRL can consistently outperform other methods with clear improvements across all statistical measures. Thanks a lot for your comment.
>
> > Q7: In the definition of $\mathcal{L}^{rec}$, what is the range of $q_{\phi}\left(\cdot \mid \theta_{k}\right) ? S ?$ Writing the expectation subscript as $s_{t, k} \sim q_{\phi}\left(\cdot \mid \theta_{k}\right)$ would help clarify things.
>
> Thanks for the suggestion. It has been updated  according to your suggestion.
>
> > Q8: In the last sentence of the Modified Cartpole setting, does $\theta_{k}=\theta_{k}^{o}$ mean the only variations are in observations?
>
> Yes, indeed.
>
> >Q9: In page 8, right above the Baselines section, it says $r_{t}=\frac{kL}{d+3L} .$ Is $k$ the actual index used in the computation of $r_{t} ?$ If not, use a different letter.
>
> Thanks for your careful reading. It is not. We have changed it to $\alpha$ to avoid possible confusion.
>
> >Q10: The authors use the word "parsimonious" a lot, which seems strange. The Oxford Dictionary says its definition is "extremely unwilling to spend money", which seems like a strange adjective to apply to an RL algorithm.
>
> Parsimonious and parsimony refer to being economical, sparing, thrifty, which is also often used in science in relation to the simplest explanation that fits the data, i.e., Occam's razor (see https://www.merriam-webster.com/dictionary/parsimony, https://en.wikipedia.org/wiki/Parsimony or https://www.cambridge.org/core/books/abs/mathematics-of-signal-processing/parsimonious-representation-of-data/).Thus, this fits particularly well with our setting, since our representation is the simplest (sparsest in terms of structure, as we encode in the regularization terms) that fits the data well, and it is a standard descriptor in model selection based on Occam's razor.
>
> > Q11: In the Introduction the authors should also reference Contrastive Behavioral Similarity Embeddings for Generalization in Reinforcement Learning. I know it is referenced later in the paper, but it should also be discussed in the introduction when other representation learning methods (such as (Zhang et al. 2021)) are being discussed.
>
> We have added it to the introduction in the updated version. Thanks for the suggestion.
>
> > Q12: At the bottom of page two it says "We learn an optimal policy", but to be precise it should be "a near-optimal policy".
>
> Thanks for the suggestion; we added a footnote "We consider optimality of the policy w.r.t. the model estimated on source domains and AdaRL assumptions."
>
> > Q13: Various Typos
>
> Thanks for your careful reading. We have fixed the typos and made the writing clearer.

---

> ### Comment · Reviewer_6qyY · 2021-11-24
> **Reviewer response**
>
> Thank you for your responses! Your rebuttal has clarified all my main concerns, so I will be raising my score.

---

### Official Review · Reviewer_awgy · 2021-11-03

**Correctness:** 3
**Technical Novelty And Significance:** 3
**Empirical Novelty And Significance:** 3
**Recommendation:** 8
**Confidence:** 4

**Main Review:**

Pros:
- I think the explicit factorization is great and that the results are impressive.
- The ablation on the masks is good and helps demonstrate that all the components of the method are needed.

Improvements and questions:
- The particular architecture i.e. the masks and factorization, seem like they might be broadly
- What happens if you get the number of parameters that vary between domains wrong i.e. you say there are two but there are actually three or vice versa?
- How does this scale with the number of source domains?

Clarity:
- In the evaluation section, you have $N_\text{target}=20, 50,10,000$ and I am pretty sure you mean $20$ and $50$ and $10000$ but it looks a little confusing. You might want to remove the comma from the 10000.
- In evaluation case, you state the you are evaluating in the POMDP case since the inputs are high dimensional. Are you stacking frames? If so, this isn’t necessarily a POMDP. (TODO check appendix)
- It seems odd to call this method interpretable. It seems to be interpretable because you know what the factors of variation are a-priori; would it still be interpretable if you didn’t? As in, does your method allow you to post-de-facto construct the factors of variation by examining the masks or something fo the sort?
- On page 2 you state that the noise factor does not affect the optimal policy for rotated + white noise pong? Is this obvious? It seems possible to construct scenarios where the noise factor very much affects the optimal policy. For example, one could imagine a game of adversarial pong where collision with the ball receives a huge negative reward but the opposing agent in pong stays the same. In such a case, it might be optimal to hide in one of the corners when the noise is high and you frequently can’t see your paddle; in the case where the noise is low you might want to actively dodge the ball. Perhaps I’m missing something and it’s obvious that noise doesn’t change the optimal policy.
- On page 2 in the final paragraph you use the notation s_{2,t} but this $2$ index is not defined at this point in time.
- Am I correct in observing the Algorithm 1 does not contain the actual inference procedure of the target value of $\theta$? Where in the main text is the inference procedure discussed?


**Summary Of The Paper:**

Instead of implicitly updating the policy using data from the source domain, learn a particularly structured latent model and the elements of variation and learn a policy that performs pretty well on some set of the elements of variation. At test time, estimate the elements of variation and provide them as input to the policy. I think this is a good paper but could benefit from some clearer writing as discussed in the clarity section below.

**Summary Of The Review:**

Small issues with the writing but the ideas seem new and to work quite well!

---

> ### Author Response · Authors · 2021-11-19
> **Thanks for the review - some additional information (part 2)**
>
> > Q6: It seems odd to call this method interpretable. It seems to be interpretable because you know what the factors of variation are a priori; would it still be interpretable if you didn’t? As in, does your method allow you to post-de-facto construct the factors of variation by examining the masks or something fo the sort?
>
> We have made ``interpretable" more specific in the updated version (see the second paragraph in Section 3.2).The factors of variation are not known as a priori;  we even have to learn whether they are affecting the observation, transition, or reward function. We learn what and where the factors of variations are, by estimating change factors $\theta$ and masks $\mathbf{c}$ from sequences of observations $\{\langle o_t, a_t, r_t  \rangle\}_{t=1}^T$.
>
> Specifically, suppose that we have three domains $i=1,2,3$ and that the estimated $\theta_i$ values in each domain satisfy the following condition: $\theta_1^o = \theta_2^o = \theta_3^o$, $\theta_1^s > \theta_2^s > \theta_3^s$, and $\theta_1^r = \theta_2^r = \theta_3^r$. Then it means that the changes are in the transition dynamics. Furthermore, with the help of the structural matrix $\mathbf{c}$, we can determine which dimensions of the state vector have changes. Since the factors of variation are estimated with our method, we claim it is interpretable in this specific sense.
>
> Figure A5 in the Appendix shows the estimated $\theta$ values in Cartpole. We can see that in the case of gravity change, the estimated $\theta_s$ is proportional to the gravity, and that in the case of mass change, the estimated $\theta_s$ is proportional to the square root of the mass. Moreover, in the change of noise level, the estimated $\theta_o$ is proportional to the square root of the added noise variance. These estimation results are reasonable, because in the underlying physical process, the gravity linearly affects the transition dynamics, while it is the square root of the mass that affects the transition dynamics. In Figure A9 in the appendix, we show that in Pong the estimated $\theta$ values are inversely proportional (in the case of varying sizes) or proportional (in the case of varying noise variances) to the factor of variation. Similarly, Figures A10 and A11 show the estimated $\theta^r$ values when the reward function changes across domains, which also reflect the underlying change of the reward function.
>
> > Q7: On page 2 you state that the noise factor does not affect the optimal policy for rotated + white noise pong? Is this obvious? It seems possible to construct scenarios where the noise factor very much affects the optimal policy. For example, one could imagine a game of adversarial pong where collision with the ball receives a huge negative reward but the opposing agent in pong stays the same. In such a case, it might be optimal to hide in one of the corners when the noise is high and you frequently can’t see your paddle; in the case where the noise is low you might want to actively dodge the ball. Perhaps I’m missing something and it’s obvious that noise doesn’t change the optimal policy.
>
> Thank you for raising this interesting point. From the perspective of generative environment models, directly adding noise to the observational images only changes the observation function, whose change can be characterized by $\theta_o$. Moreover, from the graphical illustration given in Figure 1, we can see that $\theta_o$ is independent of $r_t$, while $\theta_o$ is dependent on $r_t$ given $o_t$. It means that if we directly learn the policy from the observed image $o_t$ (i.e., the policy is a function of the observed image), $\theta_o$ will affect the policy. In contrast, if we can infer the underlying state dynamics and learn the policy as a function of the states, then $\theta_o$ will not affect the policy. Hence, if in source domains, there is at least one source domain with low noise level, then we can well learn the generative environment model, including the state dynamics. Moreover, in the target domain, even if the noise level is very high, we can still learn the optimal policy. It is because the policy is a function of the underlying states, which do not change across domains by adding noises to images.
>
> > Q8: On page 2 in the final paragraph you use the notation $s_{2,t}$ but this index $2$ is not defined at this point in time.
>
> Thanks for your careful reading. We have updated the context in the updated manuscript.
>
> > Q9: Am I correct in observing the Algorithm 1 does not contain the actual inference procedure of the target value of $\theta$? Where in the main text is the inference procedure discussed?
>
> You are correct. Algorithm 1 was originally focused on model learning and policy learning in source domains. The inference procedure is briefly discussed in the last paragraph of Section 3.1. We have updated Algorithm 1 to involve the inference procedure of the target $\theta$ as well. Thank you for your suggestion.

---

### Official Review · Reviewer_DmF1 · 2021-11-05

**Correctness:** 4
**Technical Novelty And Significance:** 3
**Empirical Novelty And Significance:** 3
**Recommendation:** 8
**Confidence:** 3

**Main Review:**

The paper is proposing an interesting method for an important problem. Transfer with minial number of steps (and no training) is a very valuable aspect for reinforcement learning. The authors do a great job at presenting the problem and the current approaches to the problem, giving enough context to the reader.

The proposed method is novel, it has significant differences compared to previous methods. The authors explain the rationale and the theory behind it in details.  For evaluation, the authors give a pretty detailed comparison with different types of target domains and through comparison with other methods. On the other hand, the selection of domains could be more complex such as locomotion environments where contact dynamics is a primary concern and transfer learning is a natural problem to tackle. Or the authors could also test changing dynamics of the pong game in addition to the representation. The results show that the AdaRL outperform other methods, but the error bounds are very large (both for AdaRL and other methods) with respect to the difference between the methods.

The writing and the presentation of the paper is very well. The paper is easy to follow and understand.
Few small details: When the authors write 10000 as 10,000 it causes weird side effects such as Ntarget = {20,50,10,000}. This should be corrected.I believe that in page 7,  0^k_o should be 0^o_k.


**Summary Of The Paper:**

The paper proposes the algorithm "AdaRL" a transfer method for reinforcement learning for different domains. The method is based on learning a latent representation with "domain shared" and "domain specific" components. A policy that is parameterized by "domain specific" components is learned. The transfer is done by collecting some data in target domain and estimating "domain specific" variable of the target domain. The authors show the algorithms mechanism for POMDP and MDP settings. The method is evaluated on modified versions of Cart-Pole and Pong domains. The authors test different settings where parameters of the environment as well as rewards functions are changed within source domains and the target domain. The authors also evaluate interpolation vs extrapolation cases. The method is compared with latest transfer learning methods and the results show that AdaRL performs better compared to other methods

**Summary Of The Review:**

The paper proposes a complex and interesting approach to transfer for RL. The authors do a throughout comparison and analysis of the method. The evaluated domains could be more complex, otherwise the results support the claims of the authors.

---

### Decision · Program_Chairs · 2022-01-20

**Decision:**

Accept (Spotlight)

**Comment:**

The authors present a method called "AdaRL" that learns a structured latent representation that characterizes relationships between different variables in an RL system. The method is evaluated on modified Pong and Cart-Pole domains and it is shown to outperform other transfer learning baselines. The reviewers agree that the method makes sense and addresses an important problem of transfer in RL. The authors did a good job in the rebuttal to empirically validate their claims and provided extra experiments. The reviewers also point out that the evaluated domains are rather simple and the paper would benefit from evaluations in a more complex environment as well as better writing. Please focus on improving these aspects in the final version of the paper.